# Look Beneath the Surface: Exploiting Fundamental Symmetry for Sample-Efficient Offline RL

**Peng Cheng**[* 1,3], **Xianyuan Zhan**[* † 2,4], **Zhihao Wu**[† 1,3], **Wenjia Zhang**[2],
**Shoucheng Song**[1,3], **Han Wang**[1,3], **Youfang Lin**[1,3], **Li Jiang**[2]

[1] Beijing Jiaotong University, Beijing, China
[2] Tsinghua University, Beijing, China
[3] Beijing Key Laboratory of Traffic Data Analysis and Mining, Beijing, China
[4] Shanghai Artificial Intelligence Laboratory, Shanghai, China
pcheng6@126.com, zhanxianyuan@air.tsinghua.edu.cn,
zhwu, yflin@bjtu.edu.cn

## Abstract

Offline reinforcement learning (RL) offers an appealing approach to real-world tasks by learning policies from pre-collected datasets without interacting with the environment. However, the performance of existing offline RL algorithms heavily depends on the scale and state-action space coverage of datasets. Real-world data collection is often expensive and uncontrollable, leading to small and narrowly covered datasets and posing significant challenges for practical deployments of offline RL. In this paper, we provide a new insight that leveraging the fundamental symmetry of system dynamics can substantially enhance offline RL performance under small datasets. Specifically, we propose a Time-reversal symmetry (T-symmetry) enforced Dynamics Model (TDM), which establishes consistency between a pair of forward and reverse latent dynamics. TDM provides both well-behaved representations for small datasets and a new reliability measure for OOD samples based on compliance with the T-symmetry. These can be readily used to construct a new offline RL algorithm (TSRL) with less conservative policy constraints and a reliable latent space data augmentation procedure. Based on extensive experiments, we find TSRL achieves great performance on small benchmark datasets with as few as 1% of the original samples, which significantly outperforms the recent offline RL algorithms in terms of data efficiency and generalizability. Code is available at: https://github.com/pcheng2/TSRL

## 1 Introduction

The recently emerged offline reinforcement learning (RL) provides a new paradigm to learn policies from pre-collected datasets without the need to interact with the environments [1, 2, 3]. This is particularly desirable for solving practical tasks, as interacting with real-world systems can be costly or risky, and high-fidelity simulators are also hard to build [4]. However, existing offline RL methods have high requirements on the size and quality of offline datasets in order to achieve reasonable performance. When such requirements are not met, these algorithms may suffer from severe performance drop, as illustrated in Figure 1. Current offline RL algorithms are trained and validated on benchmark datasets (e.g., D4RL [5]) that contain millions of transitions for simple tasks. Whereas under realistic settings, it is often impractical or costly to collect such a large amount of data, and the real datasets might only narrowly cover the state-action space. Clearly, learning reliable policies from

---

[*]Equal contribution.
[†]Corresponding Author.

37th Conference on Neural Information Processing Systems (NeurIPS 2023).

small datasets with partial coverage has become one of the most pressing challenges for successful real-world deployments of offline RL.

Unfortunately, sample-efficient design considerations have been largely overlooked in the majority of offline RL studies. Pessimism is universally adopted in existing offline RL methods and various forms of data-related regularizations have been applied to combat the distributional shift and exploitation error accumulation issues [3, 6], such as conservatively restricting policy deviation from the behavioral data [3, 6, 7, 8], regularizing value function on out-of-distribution (OOD) samples [9, 10, 11, 12], learning policy on a pessimistic MDP [4, 13, 14],

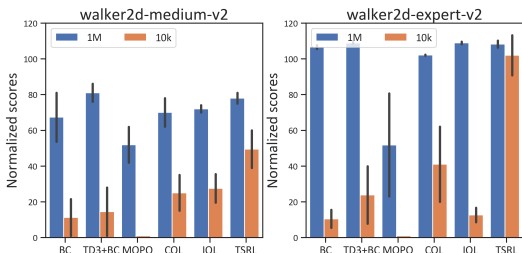

Figure 1: Performance of recent offline RL methods and our proposed TSRL on the D4RL-MuJoCo Walker2d medium and expert datasets when reducing the number of samples from 1M (full dataset) to 10k (1%).

or adopting strict in-sample learning [15, 16, 17, 18]. In many of these methods, the full coverage assumption plays an important role in their theoretical performance guarantees [3, 19], which assumes the dataset to contain all state-action pairs in the induced distribution of the policy. Obviously, most of the state-action space will become OOD regions under a small dataset. Applying strict data-related regularizations will inevitably cause severe performance degradation and poor generalization [20]. Consequently, it is important to rethink what is essential in policy learning with small datasets. In other words, what is the fundamental or invariant information that can be used to facilitate policy learning, without being conservatively confined by the limited data?

In this paper, we provide a new insight that exploiting the fundamental symmetries in the system dynamics can substantially enhance the performance of offline RL with small datasets. Specifically, we consider the time-reversal symmetry (also called *T-symmetry*), which is one of the most fundamental properties discovered in classical and quantum mechanics [21, 22]. It suggests that the underlying laws of physics should not change under the time-reversal transformation: $t \rightarrow -t$ [22, 23]. Specifically, we are interested in an extended form of T-symmetry for MDP due to its simplicity and universality in physical systems. In the small dataset setting, enforcing T-symmetry in dynamics model learning and offline RL offers three crucial benefits. First, as T-symmetry describes a fundamental property of a system, hence learning it in principle would not require a large or high-coverage dataset. Second, T-symmetry captures what is essential and invariant in the dynamical system, thus offering great generalizability. Lastly, compliance with T-symmetry also provides an important clue to detecting unreliable or non-generalizable state-action samples.

Based on these intuitions, we develop a physics-informed T-symmetry enforced Dynamics Model (*TDM*) to learn a well-behaved and generalizable dynamics model with small datasets. TDM enforces the extended T-symmetry between a pair of latent space forward and reverse dynamics sub-models, which are modeled as first-order ordinary differential equation (ODE) systems to extract fundamental dynamics patterns in data. TDM provides both well-behaved representations for small datasets and a new reliability measure for OOD samples based on compliance with the T-symmetry. These can be used to construct a highly data-efficient offline RL algorithm, which we call T-Symmetry regularized offline RL (*TSRL*). Specifically, TSRL uses the T-symmetry regularized representations learned in TDM to facilitate value function learning. Furthermore, the deviation on latent actions and the consistency with T-symmetry specified in TDM provide another perspective to detect unreliable or non-generalizable samples, which can serve as a new set of policy constraints to replace the highly restrictive OOD regularizations in existing offline RL algorithms. Lastly, a reliable latent space data augmentation scheme based on compliance with the T-symmetry is also applied to further remedy the limited size of training data. With these designs, TSRL performs surprisingly well compared with the state-of-the-art offline RL algorithms on reduced-size D4RL benchmark datasets with even as few as 1% of the original samples. To the best of the authors' knowledge, this is the first offline RL method that demonstrates promising performance on extremely small datasets.

## 2 Preliminaries

**Offline reinforcement learning.** We consider the standard Markov decision process (MDP) setting [24], which is represented as a tuple $\mathcal{M} = \{\mathcal{S}, \mathcal{A}, r, \mathcal{P}, \rho, \gamma\}$, where $\mathcal{S}$ and $\mathcal{A}$ are the state and

action spaces, $r(s, a)$ is a scalar reward function, $\mathcal{P}$ is the transition dynamics, $\rho$ is the initial state distribution, and $\gamma \in (0, 1)$ is a discount factor. The objective of RL is to learn a policy $\pi(a|s)$ by maximizing the expected cumulative discounted return $\mathbb{E}_\pi[\sum_{t=0}^{\infty} \gamma^t r(s_t, a_t))]$, which is typically approximated by a value function $Q(s, a)$ using some function approximators, such as deep neural networks. The Q-function is typically learned by minimizing the squared Bellman error:

$$Q = \arg\min_Q \mathbb{E}\left[(Q(s, a) - \mathcal{B}^\pi \hat{Q}(s, a))^2\right] \tag{1}$$

where $\hat{Q}$ denotes a target Q-function, which is a delayed copy of the current Q-function; $\mathcal{B}^\pi$ is the Bellman operator, which is often used as the Bellman evaluation operator $\mathcal{B}^\pi \hat{Q}(s, a) = r(s, a) + \gamma \mathbb{E}_{a' \sim \pi} \hat{Q}(s', a')$ in many RL algorithms.

Under the offline RL setting, we are provided with a fixed dataset $\mathcal{D} = \{(s_0, a_0, r_0, s_1, \cdots)^{(i)}\}_{i=1}^N$ without any chance of further environment interactions. Directly applying standard online RL methods in the offline setting suffers from severe value overestimation, due to counterfactual queries on OOD data and the resulting extrapolation errors [1, 3, 6]. To avoid this issue, a widely used offline RL framework adopts the following behavior regularization scheme which regularizes the divergence between the learned policy $\pi$ and the behavior policy $\pi_\beta$ of the dataset $\mathcal{D}$:

$$\pi = \arg\max_\pi \mathbb{E}_{s \sim \mathcal{D}, a \sim \pi(\cdot|s)} [Q(s, a) - D(\pi(\cdot \mid s) \| \pi_\beta(\cdot \mid s))] \tag{2}$$

where $D(\cdot \| \cdot)$ is some divergence measures, which can have either an explicit [3, 8] or implicit form [6, 17, 25]. Although straightforward, existing behavior regularization methods have been shown to be over-conservative [9, 20] due to the restrictive regularization with respect to the behavior policy in data, which may suffer from notable performance drop under small datasets.

**Time-reversal symmetry in dynamical systems.** Most real-world dynamical systems with state measurement $\mathbf{x} \in \Omega$ on some phase space $\Omega$ can be modeled or approximated by the system of non-linear first-order ordinary differential equations (ODEs) as $\frac{d\mathbf{x}}{dt} = F(\mathbf{x})$, where $F$ is some general non-linear, at least $\mathcal{C}^1$-differentiable vector-valued function. First-order ODE systems are said to be time-reversal symmetric if there is an invertible transformation $\Gamma : \Omega \mapsto \Omega$, that reverses the direction of time [22, 26]: $d\Gamma(\mathbf{x})/dt = -F(\Gamma(\mathbf{x}))$. If we define a time evolution operator $U_{\Delta t} : \Omega \mapsto \Omega$ as $U_{\Delta t} : \mathbf{x}(t) \mapsto U_{\Delta t}(\mathbf{x}(t)) = \mathbf{x}(t + \Delta t)$. Then T-symmetry implies that $\Gamma \circ U_\tau = U_{-\tau} \circ \Gamma$. In other words, the reversing of the forward time evolution of an arbitrary state should be equal to the backward time evolution of the reversed state.

**Extending T-symmetry for more generic MDP settings.** In our discrete-time MDP setting, we have $\mathbf{x} = (s, a)$. We can slightly abuse the notations and denote $\dot{s} = \frac{ds}{dt}$ as the time-derivative of the current state $s$, which can be approximated as the difference between the next and current states, i.e., $\dot{s} = s' - s$. For a dynamical system that satisfies T-symmetry, it suggests that if we learn a forward dynamics $F(s, a) = \dot{s}$ and a reverse dynamics $\tilde{G}(s', a') = -\dot{s}$ as a pair of first-order ODEs, we should have $F(s, a) = -\tilde{G}(s', a')$.

However, from a decision-making perspective, it is known that T-symmetry can sometimes be broken by irreversible actions or some special dynamic processes (e.g., frictional force against motion). Hence in this paper, we consider a more generic treatment by leveraging an alternative ODE reverse dynamics model $G(s', a) = -\dot{s}$ to establish the T-symmetry with the forward dynamics, i.e., enforcing $F(s, a) = -G(s', a)$. Note that $G(s', a)$ is now defined on the next state $s'$ and the current action $a$, rather than the next action $a'$, thus is not impacted if the next action is irreversible. This *extended T-symmetry* provides a more fundamental and almost universally held property in discrete-time MDP systems. Its simplicity and fundamentalness make it an ideal property that we can leverage to construct a well-behaved data-driven dynamics model and a robust offline RL algorithm under small datasets.

## 3 T-Symmetry Enforced Dynamics Model

In this section, we present the detailed design of TDM, which is capable of learning a more fundamental and T-symmetry preserving dynamics from small datasets. The key ingredients of TDM are to embed a pair of latent forward and reverse dynamics as ODE systems, and further enforce their T-symmetry consistency. This design offers several benefits. First, embedding ODE modeling

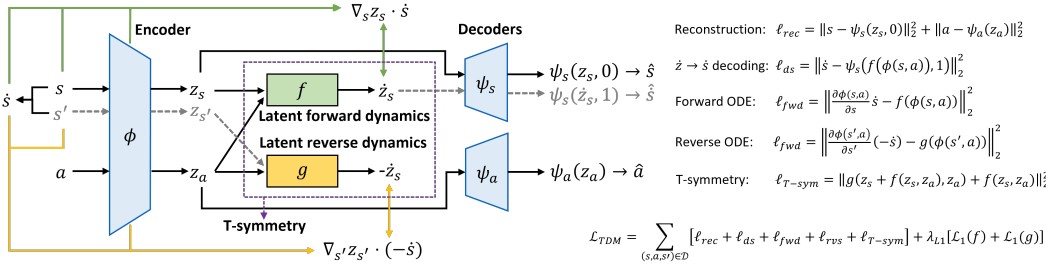

Figure 2: Overall architecture of the proposed TDM

of the system helps to extract more fundamental descriptions of how the system dynamics evolve over time in the latent space. Similar ideas have also been explored in Koopman theory [27, 28] and discovering governing equations of nonlinear dynamical systems [29, 30]. The requirement on extended T-symmetry for both forward and reverse dynamics introduces additional consistency regularization, which is helpful to improve model learning stability under limited data. With a regularized and well-behaved dynamics characterization, we can learn more effective state-action representations, which allows removing non-essential elements in the raw data to promote generalization performance. Lastly, compliance with T-symmetry can provide extra information regarding the reliability of OOD samples, which can be particularly useful in downstream offline policy learning. As illustrated in Figure 2, the proposed TDM consists the following components:

**Encoder and decoders.** TDM implements a state-action encoder $\phi(s, a) = (z_s, z_a)$ and a pair of decoders $\psi_s(\cdot, \delta_s), \psi_a(z_a) = a$ that embed the state-action pair $(s, a)$ into latent representations $(z_s, z_a)$ and then map them back. Specifically, we require the state decoder $\psi_s(\cdot, \delta_s)$ to be capable of decoding both $z_s$ and $\dot{z}_s$, where $\delta_s$ is an indicator to help the decoder to decide the target output, with $\delta_s = 0$ as decoding $z_s \to s$ and $\delta_s = 1$ as decoding $\dot{z}_s \to \dot{s}$. The encoder $\phi$ and decoders $\psi_s$ and $\psi_a$ induce the following reconstruction loss term for each state-action pair $(s, a)$:

$$\ell_{rec}(s, a) = \|s - \psi_s(z_s, 0)\|_2^2 + \|a - \psi_a(z_a)\|_2^2 \tag{3}$$

**Latent forward dynamics.** Inspired by the prior works that incorporate physics-informed information into dynamical systems modeling [27, 29, 30], we embed a discrete-time first-order ODE system to capture the latent forward dynamics $f(z_s, z_a) = \dot{z}_s$. Similar to $\dot{s}$, we write $\dot{z}_s = z_{s'} - z_s$ to denote the forward difference of the next and current latent state representations. Note that based on the chain-rule, we have $\dot{z}_s = \frac{dz_s}{dt} = \frac{\partial z_s}{\partial s} \cdot \frac{ds}{dt} = \nabla_s z_s \cdot \dot{s}$. To enforce the ODE property, we can introduce the following loss term for $f$:

$$\ell_{fwd}(s, a, s') = \|(\nabla_s z_s)\dot{s} - \dot{z}_s\|_2^2 = \|\frac{\partial \phi(s, a)}{\partial s}\dot{s} - f(\phi(s, a))\|_2^2 \tag{4}$$

Minimizing $\mathcal{L}_{fwd}$ ensures that the latent forward dynamics $f$ correctly predicts the forward time evolution of latent states in the dynamical system. We also require the decoder $\psi_s(\cdot, \delta_{\dot{s}})$ to be able to decode $\dot{s}$ from $\dot{z}_s$ to ensure it is compatible with the ODE property, which implies the following loss:

$$\ell_{ds}(s, a, s') = \|\dot{s} - \psi_s(\dot{z}_s, 1)\|_2^2 = \|\dot{s} - \psi_s(f(\phi(s, a)), 1)\|_2^2 \tag{5}$$

**Latent reverse dynamics.** We can further introduce a latent reverse dynamics $g(z_{s'}, z_a) = -\dot{z}_s$ in the model, which captures the reverse time evolution of the system in the latent space. Similar to the forward dynamics loss $\mathcal{L}_{fwd}$, we can write the reverse dynamics loss for $g$ as:

$$\ell_{rvs}(s, a, s') = \|(\nabla_{s'} z_{s'})(-\dot{s}) - (-\dot{z}_s)\|_2^2 = \|\frac{\partial \phi(s', a)}{\partial s'}(-\dot{s}) - g(\phi(s', a))\|_2^2 \tag{6}$$

**T-symmetry regularization.** The above latent forward and reverse dynamics $f$ and $g$ are learned to be two models, which may not necessarily satisfy the proposed extended T-symmetry. We can enforce the extended T-symmetry by requiring $f(z_s, z_a) = -g(z_{s'}, z_a)$. To further couple the learning process of $f$ and $g$, note that $z_{s'} = z_s + \dot{z}_s = z_s + f(z_s, z_a)$, which suggests $g(z_{s'}, z_a) = g(z_s + f(z_s, z_a), z_a) = -\dot{z}_s = -f(z_s, z_a)$. This implies the following T-symmetry consistency loss:

$$\ell_{T-sym}(z_s, z_a) = \|f(z_s, z_a) + g(z_s + f(z_s, z_a), z_a)\|_2^2 \tag{7}$$

Above instance-wise T-symmetry consistency loss also provides an alternative measure for evaluating the reliability of a data sample. A state-action pair $(s, a)$ with a large $\ell_{T\text{-}sym}(\phi(s, a))$ implies that this sample may not be well-explained by TDM or consistent with the fundamental symmetry of the system. This can be used to detect unreliable OOD samples in offline policy optimization as well as construct a new latent space data augmentation procedure, which will be discussed in later content.

**Final learning objective of TDM.**    Finally, we can formulate the overall loss function of TDM as:

$$\mathcal{L}_{TDM} = \sum_{(s,a,s') \in \mathcal{D}} [\ell_{rec} + \ell_{ds} + \ell_{fwd} + \ell_{rvs} + \ell_{T\text{-}sym}](s, a, s') + \lambda_{L1}[\mathcal{L}_1(f) + \mathcal{L}_1(g)] \tag{8}$$

where $\mathcal{L}_1(f)$ and $\mathcal{L}_1(g)$ are L1-norms of the parameters of $f$ and $g$, and $\lambda_{L1}$ is a scale parameter. L1 regularization is introduced to encourage learning parsimonious latent dynamics for $f$ and $g$, which helps to improve model generalizability [29, 30].

Note that the proposed TDM is very different from the conventional dynamics models used in model-based RL (MBRL) methods [4, 13, 14, 31, 32, 33, 34, 35]. The dynamics models in MBRL focus on constructing a predictive model to represent the forward transition dynamics of the system. Whereas, TDM is formulated as a reconstruction model with T-symmetry preserving embedded ODE latent dynamics, which aims at explaining and extracting the fundamental dynamics of the system. As a result, TDM can be substantially more well-behaved and robust when learning from small datasets.

## 4   T-Symmetry Regularized Offline RL

In this section, we discuss how to incorporate the properties of TDM to construct a sample-efficient offline RL algorithm, which we call T-Symmetry regularized offline RL (TSRL). Specifically, we leverage three types of information from TDM to jointly improve offline RL performance under small datasets. First, we use the well-behaved state-action representations provided by TDM to facilitate value function learning. Second, the consistency with the latent dynamics in TDM also provides new forms of policy constraints to penalize unreliable and non-generalizable OOD samples. Finally, compliance with T-symmetry enables a new latent space data augmentation procedure to further enhance the algorithm's performance with limited data.

**T-symmetry regularized representation.**    Representation learning has been shown to be an effective approach to enhancing sample efficiency and generalization in many online and offline RL studies [36, 37, 38, 39, 40]. A notable property of TDM is that the learned latent state-action representations from the encoder $(z_s, z_a) = \phi(s, a)$ are compatible with both the latent forward and reverse ODE dynamics $f$ and $g$. This leads to well-regularized and T-symmetry preserving representations that can potentially generalize better on OOD areas under small dataset settings. We can simply use the latent state-action representation $(z_s, z_a)$ extracted by the encoder $\phi(s, a)$ of TDM in the value function learning, which gives the following policy evaluation objective:

$$Q = \underset{Q}{\arg\min} \, \mathbb{E}_{(s,a,s') \sim \mathcal{D}} \left[ \left( r(s, a) + \gamma \hat{Q}(\phi(s', \pi(\cdot|s'))) - Q(\phi(s, a)) \right)^2 \right] \tag{9}$$

**T-symmetry regularized policy constraints.**    Existing offline RL methods primarily penalize the divergence between the learned policy $\pi$ and the behavioral data in the original action space, which ignores the underlying manifold structure of actions in the latent space [41] and the system dynamics properties. Moreover, restricting policy optimization only within data-covered regions can be over-conservative when the offline dataset is small, which can result in degraded policy performance [20]. In TSRL, we instead consider an alternative regularization scheme, which restricts the deviation on latent actions and the T-symmetry consistency of policy-induced samples, corresponding to the following policy optimization objective:

$$\underset{\pi}{\arg\max} \, \mathbb{E}_{(s,a) \sim \mathcal{D}} \left[ \alpha Q(\phi(s, \pi(\cdot|s))) - \lambda_1 \| z_{a^\pi} - z_a \|_2^2 - \lambda_2 \ell_{T\text{-}sym}(\phi(s, \pi(\cdot|s))) \right] \tag{10}$$

where latent actions $z_a$ and $z_{a^\pi}$ are obtained from $\phi(s, a)$ and $\phi(s, \pi(\cdot|s))$ respectively. The second term restricts the latent action $z_{a^\pi}$ of policy $\pi$ from deviating too much from the latent action $z_a$ in data. The third term regularizes the T-symmetry consistency of policy-induced samples $(s, \pi(\cdot|s))$, which is evaluated based on Eq. (7) and the learned TDM. $\lambda_1$ and $\lambda_2$ are weight parameters, which only need to be roughly adjusted to ensure both the regularization terms are in a similar scale

as the first term. We also introduce a normalization term $\alpha$ on the value function for training stability similar to TD3+BC [8], which is computed based on a training batch $B$ of samples as $\alpha_0 / [\sum_{(s,a) \in B} Q(\phi(s, \pi(\cdot|s)))]$. We set $\alpha_0 = 2.5$ in all of our experiments without tuning.

Instead of strictly regularizing OOD actions as in existing offline RL algorithms, our policy regularization scheme actually allows some reliable OOD action for policy optimization. As TDM is designed to capture the fundamental and invariant system dynamics patterns in the latent space, if an OOD action has a similar latent representation to some latent actions in data and also agrees with the T-symmetry property in TDM, then we can expect some degree of equivalency between these actions. This leads to more relaxed policy constraints by enabling policy learning and generalization on reliable OOD regions, which is critical for the small dataset setting.

**T-symmetry consistent latent space data augmentation.**    It has been shown in previous studies [42, 28, 43] that data augmentation can potentially improve the function approximation of the Q-networks by smoothing out the learned state-action space, hence often lead to more robust policy and better data efficiency. However, existing data augmentation methods in offline RL studies either blindly add random perturbations to states [42] or utilize costly non-linear symmetry transformations, such as Koopman theory [28]. With TDM, we can provide a very simple yet principled data augmentation scheme based on the T-symmetry property. Assuming we add a small perturbation $\epsilon$ to a latent state $z_s$, i.e., $(z_s, z_a) \mapsto (z_s + \epsilon, z_a)$, then the corresponding perturbation $\epsilon'$ on the next latent state $z_{s'}$ according to the latent forward dynamics $\dot{z}_s = f(z_s, z_a)$ satisfies: $z_{s'} + \epsilon' = z_s + \epsilon + f(z_s + \epsilon, z_a)$. On the other hand, by the T-symmetry construction in TDM, we can recover back the current perturbed latent state based on the latent reverse dynamics $-\dot{z}_s = g(z_{s'}, z_a)$ as: $z_s + \epsilon'' = z_{s'} + \epsilon' + g(z_{s'} + \epsilon', z_a)$. Clearly, we should have $\epsilon = \epsilon''$, which suggests the following condition:

$$\epsilon'' - \epsilon = f(z_s + \epsilon, z_a) + g(z_s + \epsilon + f(z_s + \epsilon, z_a), z_a) = 0 \tag{11}$$

This is exactly equivalent to requiring the instance-wise T-symmetry consistency loss (Eq. (7)) $\ell_{T\text{-}sym}(z_s + \epsilon, z_a) = 0$. Hence we can use T-symmetry consistency loss $\ell_{T\text{-}sym}(\cdot)$ as a reliability measure to filter out unreliable augmented samples $(z_s + \epsilon, z_a)$ that are inconsistent with the T-symmetry property of the learned latent dynamics in TDM. In our implementation, we only keep augmented samples that satisfy $\ell_{T\text{-}sym}(z_s + \epsilon, z_a) \leq h$, where we consider a non-parametric treatment for threshold $h$, by setting it as the $\tau$-quantile value of all $\ell_{T\text{-}sym}(\phi(s, a))$ values of $(s, a)$ in $\mathcal{D}$ (we choose $\tau = 50\%$ or $70\%$ in our experiments). This ensures that the augmented samples at least maintain the similar level of T-symmetry agreement explained by TDM as the data samples in $\mathcal{D}$.

**Practical implementation.**    TSRL can be implemented based upon TD3 [2] by incorporating the proposed T-symmetry regularized representation and policy constraints as in Eq. (9) and (10), as well as the T-symmetry consistent latent space data augmentation. In our experiments reported in the next section, we generate $K = 1$ augmented samples for each transition in the dataset, and filter based on the T-symmetry consistency loss. The pseudo-code of TSRL is summarized in Algorithm 1.

---

**Algorithm 1** T-Symmetry Regularized Offline RL (TSRL)

---

**Require:** Offline dataset $\mathcal{D}$, encoder $\phi$, latent forward and reverse dynamics models $f$ and $g$ from TDM trained using objective Eq. (8).
1: Compute the T-symmetry consistency loss $\ell_{T\text{-}sym}(\phi(s, a))$ (Eq. (7)) for all samples in $\mathcal{D}$, and set their $\tau$-quantile value as the augmentation threshold $h$.
2: Initialize the policy network $\pi$, critic networks $Q$ and their target network.
3: **for** $t = 1, \cdots, M$ training steps **do**
4:      Sample a mini-batch $B$ of samples $\{(s, a, r, s')\} \sim \mathcal{D}$ and compute their representations $\{(z_s, z_a, z_{s'})\}$.
5:      *// T-symmetry consistent latent space data augmentation*
6:      Generate $K$ perturbed samples by adding perturbations $\epsilon \sim N(0, 0.01\sigma_{z_s})$ on latent states $z_s$ of each sample in $B$, where $\sigma_{z_s}$ is the std of latent states in data.
7:      Add augmented samples $(z_s + \epsilon, z_a, z_{s'} + \epsilon')$ to $B$ if satisfies $\ell_{T\text{-}sym}(z_s + \epsilon, z_a) \leq h$.
8:      *// Critic training with T-symmetry regularized representation*
9:      Update the value function $Q$ based on the policy evaluation objective Eq. (9).
10:      *// Policy training with T-symmetry regularized policy constraints*
11:      Update the policy $\pi$ based on the policy improvement objective Eq. (10).
12:      Soft update the target networks.
13: **end for**

---

Table 1: Average normalized score on D4RL MuJoCo and Adroit tasks with full and reduced-size datasets. Some of the full dataset performance scores are reported from the IQL [15], MOPO [13], and DOGE [20] papers. Complete scores for Adroit-human and cloned tasks are included in Appendix C.

| Task | Ratio | Size | BC | TD3+BC | MOPO | CQL | IQL | DOGE | TSRL(ours) |
|---|---|---|---|---|---|---|---|---|---|
| Hopper-m | 1 | 1M | 52.9 | 59.3 | 28.0 | 58.5 | 66.3 | **98.6 ± 2.1** | 86.7±8.7 |
| | 1/100 | 10k | 29.7±11.7 | 40.1±18.6 | 5.5±2.3 | 43.1±24.6 | 46.7±6.5 | 44.2 ± 10.2 | **62.0±3.7** |
| Hopper-mr | 1 | 400k | 18.1 | 60.9 | 67.5 | **95.0** | 94.7 | 76.2±17.7 | 78.7±28.1 |
| | 1/40 | 10k | 12.1±5.3 | 7.3±6.1 | 6.8±0.3 | 2.3±1.9 | 13.4±3.1 | 17.9 ± 4.5 | **21.8±8.2** |
| Hopper-me | 1 | 2M | 52.5 | 98.0 | 23.7 | **105.4** | 91.5 | 102.7± 5.2 | 95.9±18.4 |
| | 1/200 | 10k | 27.8±10.7 | 17.8±7.9 | 5.8±5.8 | 29.9±4.5 | 34.3±8.7 | 50.5 ± 25.2 | **50.9±8.6** |
| Hopper-e | 1 | 1M | 108.0 | 100.1 | 16.2±6.2 | 98.4 | 99.3 | 107.4 ± 3.6 | **110.0 ±3.3** |
| | 1/100 | 10k | 20.8±6.9 | 23.2±18.2 | 6.5±3.7 | 33.0±22.2 | 38.4±11.3 | 54.5±21.5 | **82.7±21.9** |
| Halfcheetah-m | 1 | 1M | 42.6 | **48.3** | 42.3 | 44.0 | 47.4 | 45.3± 0.6 | 48.2 ±0.7 |
| | 1/100 | 10k | 26.4±7.3 | 16.4±10.2 | -1.1±4.1 | 35.8±3.8 | 29.9±0.12 | 36.2±3.4 | **38.4±3.1** |
| Halfcheetah-mr | 1 | 200k | 55.2 | 44.6 | **53.1** | 45.5 | 44.2 | 42.8 ±0.6 | 42.2±3.5 |
| | 1/20 | 10k | 14.3±7.8 | 17.9±9.5 | 11.7±5.2 | 8.1±9.4 | 22.7±6.4 | 23.4±3.6 | **28.1±3.5** |
| Halfcheetah-me | 1 | 2M | 55.2 | 90.7 | 63.3 | 91.6 | 86.7 | 78.7±8.4 | **92.0±1.6** |
| | 1/200 | 10k | 19.1±9.4 | 15.4±10.7 | -1.1±1.4 | 26.5±10.8 | 10.5±8.8 | 26.7±6.6 | **39.9±21.1** |
| Halfcheetah-e | 1 | 1M | 92.2 | 82.1 | 1.4±2.2 | **95.6** | 88.9±1.2 | 93.5 ± 4.2 | **94.3±5.5** |
| | 1/100 | 10k | 1.10±2.4 | 1.72±3.3 | -0.6±1.1 | 4.2±0.94 | -2.0±0.4 | 1.4±2.1 | **40.6±24.4** |
| Walker2d-m | 1 | 1M | 75.3 | 83.7 | 17.8 | 72.5 | 78.3 | **86.8 ± 0.8** | 77.5 ±4.5 |
| | 1/100 | 10k | 15.8±14.1 | 7.4±13.1 | 3.1±4.7 | 18.8±18.8 | 22.5±3.8 | 45.1 ± 10.2 | **49.7±10.6** |
| Walker2d-mr | 1 | 300k | 26.0 | 81.8 | 39.0 | 77.2 | 73.9 | **87.3 ± 2.3** | 66.1±12.0 |
| | 1/30 | 10k | 1.4±1.9 | 5.7±5.8 | 3.3±2.7 | 8.5±2.19 | 10.7±11.9 | 13.5± 8.4 | **26.0±11.3** |
| Walker2d-me | 1 | 2M | 107.5 | 110.1 | 44.6 | 108.8 | 109.6 | **110.4±1.5** | 109.8±3.12 |
| | 1/200 | 10k | 21.7±8.2 | 7.9±9.1 | 0.6±2.7 | 19.1±14.4 | 26.5±8.6 | 35.3 ± 11.6 | **46.4±17.4** |
| Walker2d-e | 1 | 1M | 107.9 | 108.2 | 0.1±0.3 | 101.3 | **109.7±0.1** | 107.3±2.3 | **110.2±0.3** |
| | 1/100 | 10k | 10.4±5.3 | 23.8±16.0 | 1.4±3.4 | 41.6±21.6 | 12.6±4.5 | 72.1 ±16.2 | **102.2±11.3** |
| Adroit-human-total | 1 | 5k | 36.4 | 10.6 | 9.5 | 52.2 | 77.3 | 39.0 ± 17.1 | **80.9±21.1** |
| Adroit-cloned-total | 1 | 500k | 57.5 | 41.1 | -1.2 | 41.6 | 40.8 | 56.7 ± 16.2 | **66.3±23.7** |
| | 1/50 | 10k | 29.5±37.8 | 0.2±0.1 | -1.7±1.5 | 0.6±0.8 | 32.7±24.6 | 29.7 ± 20.4 | **44.9±25.7** |

## 5 Experiments

We evaluate TSRL on the D4RL MuJoCo-v2 and Adroit-v1 benchmark datasets [5] against behavior cloning (BC) as well as state-of-the-art (SOTA) offline RL methods, including model-free methods TD3+BC [8], CQL [9], IQL [15], DOGE [20], and model-based method MOPO [13]. In particular, we compare with DOGE [20], which leverages a state-conditioned distance function as policy constraint and exhibits strong OOD generalization performance. We report the final normalized performance of each algorithm after training 1M steps.

**Performance on small datasets.** We compare the performance of TSRL and the baseline methods on both the full D4RL datasets and their reduced-size datasets with only 5k∼10k samples, which are constructed by randomly sampling a given fraction of trajectories in the full datasets*. These reduced-size datasets are only about 1/20∼1/200 of their original size. Compared with the performances on the full datasets, most baseline offline RL methods suffer from a noticeable performance drop under these extremely small datasets, mainly due to their over-reliance on the size and coverage of training data. Among all baselines, DOGE [20] is the only baseline that can still achieve reasonable performance in some tasks, primarily due to its capability to leverage the relationship between dataset geometry and the generalization capability of deep Q-functions, which is also beneficial for policy learning under small datasets. However, we still observe that DOGE struggles in some datasets, especially those with an extremely narrow distribution.

By contrast, TSRL achieves substantially better performance in all small dataset tasks, indicating superior sample efficiency. Moreover, although MOPO [13] also learns a dynamics model for offline policy learning, it performs badly when the dataset is small, revealing the importance of using a well-regularized model like TDM in the small-sample regime. It is interesting to see that most

---

*We didn't construct reduced-size datasets for Adroit-human tasks, as the full datasets are already very small.

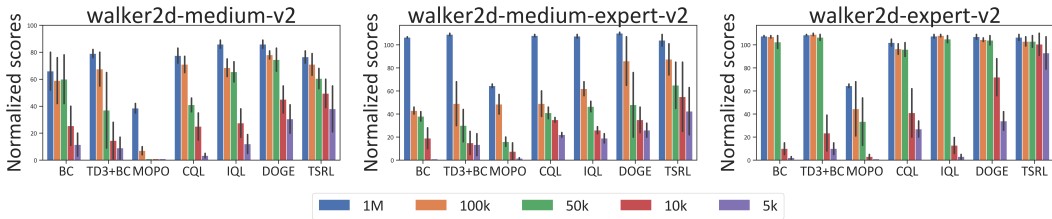

Figure 3: Performance of TSRL and baselines algorithms on datasets with different training sample sizes

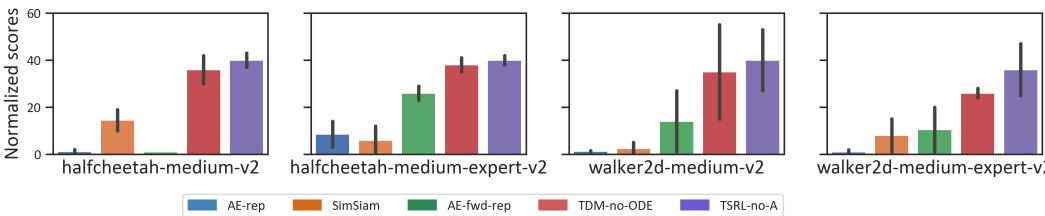

Figure 4: Impacts of different representation learning methods on 10k datasets.

methods fail on the 10k Halfcheetah-expert dataset, with only TSRL showing some success. We suspect this is due to the extremely narrow data distribution of the reduced dataset, which requires strong OOD generalization ability for algorithms to solve the task.

To further examine the impact of the training data size on algorithm performance, we also conduct experiments on three Walker2d datasets (medium, medium-expert, and expert) by varying the size of samples from 1M to 5k. The results are presented in Figure 3. It can be observed that most baseline offline RL algorithms experience a sharp performance drop when the datasets are reduced to 10k samples. Whereas, TSRL is still capable of preserving reasonable performance as the decrease of data size, even for extremely small datasets that contain only 5k samples.

**Investigation on learned representations.**   To investigate the quality of the latent representation learned in TDM, we compare the performance of different representation learning approaches on the 10k datasets in Figure 4. To solely evaluate the impact of the representation, we remove the latent space data augmentation component from TSRL ("TSRL-no-A") and replace the state-action encoder $\phi(s, a)$ learned from other representation learning approaches, including the autoencoder ("AE-rep"), autoencoder with latent forward dynamics ("AE-fwd-rep") without the ODE structure and the T-symmetry regularization in TDM, and a recent popular self-supervised representation learning method SimSiam [44] ("SimSiam"). To further investigate the impact of enforcing the ODE property, we also consider a variant of TDM ("TDM-no-ODE") by removing the ODE structure in latent forward and reverse dynamics. More detailed experiment setups are presented in Appendix B.

The results demonstrate that TDM representation achieves the best performance in all small-dataset experiments. By comparing "TSRL-no-A" and "TDM-no-ODE" with "AE-rep" and "AE-fwd-rep", we can see that the bi-directional design and the T-symmetry regularization are crucial for performance improvement. Moreover, we find "TSRL-no-A" consistently achieves better performance and lower variance as compared to "TDM-no-ODE", further confirming the benefit of incorporating ODE structure in producing a well-behaved representation for downstream tasks under small datasets.

**Evaluation on data augmentation.**   We evaluate the impact of the proposed T-symmetry consistent latent space data augmentation in Figure 5. The results show that our proposed data augmentation scheme can help speed up convergence and reduce variance. Compared with less principled data augmentation methods such as adding zero-mean Gaussian noises as in S4RL [42], our method offers much better performance improvement. As shown in Figure 5, blindly adding random perturbations could suffer from performance degradation over the course of training, while TSRL with T-symmetry consistent data augmentation enjoys better training robustness. Additional comparative experiments on the proposed data augmentation method can be found in Table 6 of Appendix C.

**Additional ablations.**   We also investigate the joint impact of the three design components in TSRL, including T-symmetry regularized representation and policy constraints, as well as the T-symmetry

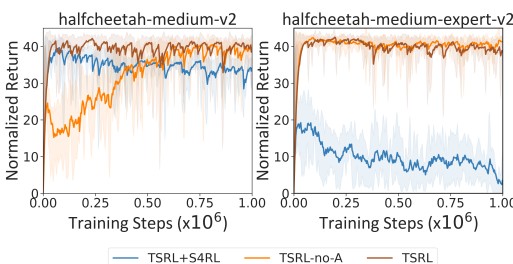
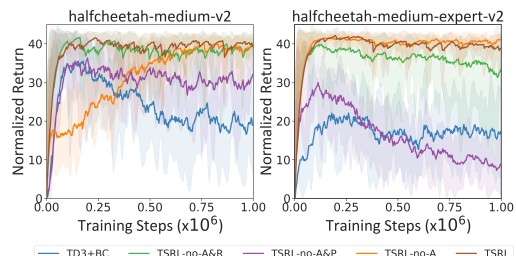

Figure 5: Impact of T-symmetry consistent data augmentation on 10k datasets. "TSRL+S4RL" denotes replacing the latent space data augmentation in TSRL with the zero-mean Gaussian noise ($N(0, 0.1\boldsymbol{I})$).

Figure 6: Ablation on TSRL on 10k datasets. "no-R": no T-symmetry regularized representation; "no-P": no T-symmetry policy constraints, and use BC term as in TD3+BC; "no-A": no latent space data augmentation.

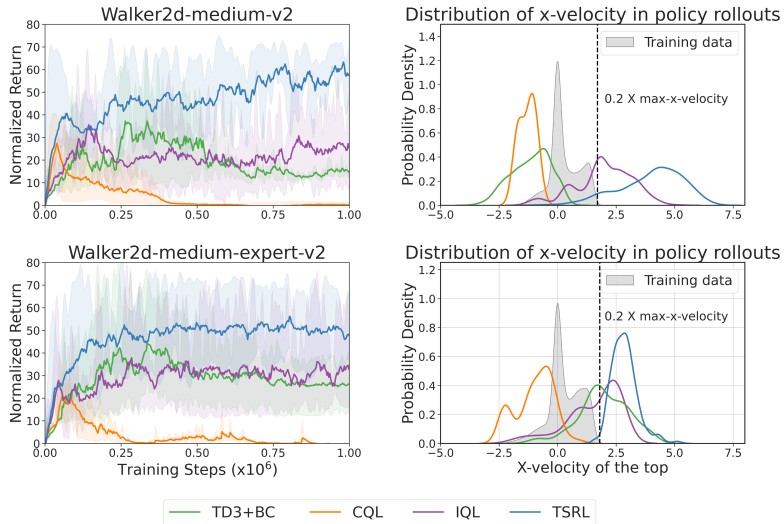

Figure 7: Comparison of TSRL and baselines trained on the Walk2d-medium (top) and medium-expert (bottom) datasets that removing all samples with x-velocity of the top $> 0.2 \times$ max-x-velocity in the data. Left: learning curves. Right: x-velocity distribution of policy evaluation rollouts during the last 10k training steps.

consistent latent space data augmentation. We include TD3+BC for comparison, as it can be perceived as a vanilla version of TSRL without the previous three components. Moreover, the variant of TSRL with only the latent representation removed is not evaluated, as the latent space data augmentation also depends on the representation provided by TDM. Figure 6 presents the performance of all variants of TSRL on the 10k Halfcheetah medium and medium-expert datasets. As expected, it is observed that T-symmetry regularized representation and policy constraints jointly play a critical role in maintaining the performance of TSRL. As discussed previously, adding T-symmetry consistent latent space data augmentation also shows a positive impact on performance. Furthermore, it is observed that TD3+BC suffers from over-fitting on small datasets as its performance drops significantly with the increase of training steps, especially in the 10k Halfcheetah-medium dataset, while this phenomenon is not observed in TSRL. Nevertheless, the complete TSRL achieves the best performance in both tasks.

**Generalization performance.** To further verify the generalizability of TSRL, we construct low-speed datasets from the Walker2d-medium and Walker2d-medium-expert datasets by filtering out all high x-velocity samples (x-velocity of the top $> 0.2 \times$ max-x-velocity). This results in two smaller datasets with a large proportion of transition dynamics unobserved. We want to test if the agent can still generalize and learn well given only these low x-velocity data, with all high-speed samples removed. The experiment results are presented in Figure 7.

Under this setting, we observed that all existing offline RL baselines perform poorly when trained with only the low-speed dataset. This is primarily due to over-conservative data-related regularizations, which cause ineffective policy learning if the OOD region occupies the majority of state-action

space. CQL performs especially poorly in both tasks, perhaps due to over-conservative value function learning that impedes the policy to acquire some necessary control strategy to finish the task. TD3+BC performs poorly in the low-speed medium dataset, likely because this dataset has a narrower data distribution than the medium-expert dataset, and the latter could still contain some samples with reasonable speeds after filtering with x-velocity$> 0.2\times$max-x-velocity in the dataset. IQL exhibits some level of generalization capability, but is still much weaker as compared to TSRL. By contrast, we observe that TSRL is still able to achieve good performance, due to the access to more fundamental dynamics information that remains invariant in both low- and high-speed data. This can be further verified if we inspect the policy rollout distributions (right figures of Figure 7) that the policies learned by TSRL indeed generalizes to high-speed behavior that is not present in the training data.

## 6 Related Work

**Learning fundamental dynamics in physical systems.** Learning conservation laws or invariant properties within a physical system is an active research area in physics [45, 46, 29, 30], climate science [47], and neuroscience [48], etc. A classic approach is based on Koopman theory, which represents the nonlinear dynamics in terms of an infinitedimensional linear operator [27]. In practice, this is achieved by finding a coordinate transformation to produce a finite-dimensional representation in which the non-linear dynamics are approximately linear. However, it also suffers from computationally expensive coordinate transformations and is only able to approximate the system dynamics. Another approach is utilizing a sparse regression model with the fewest terms to describe the nonlinear system dynamics [29, 30]. However, it assumes that the dynamical systems only have a few critical terms, which severely limits the model expressiveness and often requires prior knowledge of these critical terms. Based on expressive deep neural networks, a recently emerged research direction is to build ODE networks to learn conservation law in the dynamical system from data [49, 50, 51, 26]. Our proposed TDM falls within this direction, which models both forward and reverse latent ODE dynamics with deep neural networks and incorporates additional regularization on T-symmetry.

**Offline reinforcement learning.** Offline RL addresses the challenge of deriving policies from fixed, pre-collected datasets without interaction with the environment. Under this offline learning paradigm, conventional off-policy RL approaches are prone to substantial value overestimation when there is a large deviation between the policy and data distributions. Existing offline RL methods address this issue by following several directions, such as constraining the learned policy to be "close" to the behavior policy [6, 3, 8, 52], regularizing value function on OOD samples [9, 10, 11, 53], enforcing strict in-sample learning [16, 15, 17, 18, 54], and performing pessimistic policy learning with uncertainty-based reward or value penalties [13, 14, 4, 12, 55, 33]. Most existing offline RL methods adopt the pessimism principle and avoid policy evaluation on OOD samples. Although this treatment helps to alleviate exploitation error accumulation, it can be over-conservative and causes severe performance degradation if the training dataset is small or has poor state-action space coverage [20]. TSRL tackles this issue by allowing dynamics explainable OOD samples for policy optimization, thus offering greatly improved small-sample performance.

## 7 Discussion and Conclusion

In this paper, we propose a physics-informed dynamics model TDM and a new offline RL algorithm TSRL, which exploit the fundamental symmetries in the system dynamics for sample-efficient offline policy learning. TDM embeds and enforces T-symmetry between a pair of latent forward and reverse ODE dynamics to learn fundamental dynamics patterns in data. The well-behaved representations and a new reliability measure for OOD samples based on T-symmetry from TDM can be readily used to construct the proposed TSRL algorithm, which achieves strong performance on small D4RL benchmark datasets and exhibits good generalization ability. There are also some limitations in our proposed approach. For example, in order to learn a well-behaved dynamics model, we introduced a set of dynamics and symmetry regularizations in TDM, which are beneficial to improve model generalization, but will lose some model expressiveness. However, we believe this can be a worthwhile trade-off between precision and generalization under small dataset settings, due to substantially improved model robustness.

## Acknowledgments and Disclosure of Funding

This work is supported by National Key Research and Development Program of China under Grant (2022YFB2502904), and funding from Global Data Solutions Limited and Intel Corporation.

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

# A  Implementation Details

**Implementation details for TDM.**  As discussed in Section 3 and the illustrative Figure 2, TDM is modeled as a physics-informed reconstruction model with embedded ODE latent dynamics and T-symmetry preserving design.To ensure optimal training performance of TDM, we have included some additional implementation details below. More hyperparameters details of TDM are discussed in the next section.

- **Network structure:** In all our D4RL experiments, we implement the encoder, decoders, latent forward and reverse dynamics as 4-layer feed-forward neural networks with ReLU activation, and optimized using Adam optimizer. For the state decoder $\psi(\cdot, \delta_s)$, we concatenate an extra indicator $\delta_s$ in the input to help the state decoder to decide the target output. More specifically, to decode $z_s \rightarrow s$, we concatenate $\delta_s = 0$ with $z_s$ as input; and for $\dot{z}_s \rightarrow \dot{s}$, we concatenate $\delta_s = 1$ with $\dot{z}_s$.

- **Computing second derivative of $\phi(\cdot)$:**  As TDM involves a pair of latent ODE forward and reverse dynamics models, whose training losses Eq. (4) and (6) involve regressing on $\frac{\partial \phi(s,a)}{\partial s}\dot{s}$ and $\frac{\partial \phi(s',a)}{\partial s'}(-\dot{s})$ as target values. This results in a gradient through a gradient of $\phi(\cdot)$. Computationally, we calculate the Jacobian matrix $\frac{\partial \phi(s,a)}{\partial s}$ using the `vmap()` function in Functorch[†] to ensure the second derivative of $\phi(\cdot)$ can be correctly backpropagated during stochastic gradient descent. Similar a treatment can also be implemented with other auto-differentiation frameworks like Jax[‡] that support computing higher-order derivatives.

- **Pre-training the encoder and decoders:** As the final learning objective of TDM Eq. (8) involves several loss terms, we observe that in small datasets, loss terms such as the reconstruction loss (Eq. (3)) for the encoder and decoders converges much slower than other loss terms. When updating all the loss terms with the same number of training steps, some losses suffer from over-fitting while others are still not fully converged. For these cases, we pre-train the encoder and decoders with the reconstruction loss for a given number of training steps, and then use the complete learning objective of TDM (Eq. (4)) for the rest of the training. The numbers of pre-training/training epochs for the experiments in this paper are reported in Table 2.

  As reported in Table 2, we find that the number of pre-training epochs required for TDM to reach the best learning performance is associated with the specific task and the size of training data. For small datasets, TDM generally needs more training and pre-training epochs to avoid overfitting the latent dynamics and T-symmetry losses. For MuJoCo locomotion tasks, we recommend pre-training the encoder and decoders for 10% of the total training epoch. For the more complex adroit tasks, TDM requires more epochs to extract the ODE dynamics and T-symmetry property of the system dynamics. In this case, there is no pre-training necessary for the encoder and decoders.

Table 2: Training epochs of TDM for D4RL tasks with different dataset scales

|  | **Locomotion Tasks** | | | **Adroit Tasks** | |
|---|---|---|---|---|---|
|  | 5k&10k | 50k & 100k | Full dataset | 5k&10k | Full dataset |
| Training epoch | 2000 | 1000 | 200 | 2000 | 200 |
| Pre-train epoch | 200 | 100 | 20 | 0 | 0 |

- **Enhancement on the T-symmetry regularization:** We observe that in some small datasets (mainly in the Halfcheetah environment), the training of the latent reverse dynamics model $g$ might suffer from a certain level of degeneration. This is reflected as the $g(z_s + f(z_s, z_a), z_a)$ produces similar values as $-f(z_s, z_a)$, resulting in small T-symmetry consistency loss values (Eq. (7)), however, the discrepancy between $g(z_{s'}, z_a)$ and $-f(z_s, z_a)$ remains large. To solve this issue and further enforce the T-symmetry, we apply the following enhanced T-symmetry regularization when such a phenomenon is observed:

$$\ell_{Enhanced\text{-}T\text{-}sym}(z_s, z_a) = \|f(z_s, z_a) + g(z_s + f(z_s, z_a), z_a)\|_2^2 + \|f(z_s, z_a) + g(z_{s'}, z_a)\|_2^2 \quad (12)$$

We find that applying the above enhanced T-symmetry loss can successfully resolve the degeneration issue of the latent reverse dynamics model and achieve good performance in the downstream offline

---

[†]https://pytorch.org/functorch/stable/functorch.html
[‡]https://github.com/google/jax

RL tasks. However, we find in most small datasets, the original T-symmetry consistency loss is sufficient. We advise only to use the above enhanced T-symmetry consistency loss when large discrepancies between $\|f(z_s, z_a) + g(z_s + f(z_s, z_a), z_a)\|_2^2$ and $\|f(z_s, z_a) + g(z_{s'}, z_a)\|_2^2$ are observed.

**Hyperparameter details for TDM and TSRL.** The architectural parameters of TDM and TSRL, as well as the TSRL hyperparameters are summarized in Table 3. Based on the different scales of the datasets, we basically only use two sets of hyperparameters for all D4RL-MuJoCo locomotion tasks and only one set of hyperparameters for all D4RL-Adroit tasks. Because of the extremely narrow distribution of the reduced-size expert dataset, we apply Dropout [56] with dropout rate of 0.1 to regularize the policy network in all tasks with 10k expert data.

Table 3: Hyperparameter details for TDM and TSRL

|  | Hyperparameters | Value |
|---|---|---|
| TDM Architecture | Optimizer type | Adam |
|  | Weight of $\ell_{T-sym}$ and $\ell_{ds}$ and $\ell_{rec}$ | 1 |
|  | Weight of $\ell_{rvs}$ and $\ell_{fwd}$ | 0.1 |
|  | Learning rate | $3 \times 10^{-4}$ |
|  | State normalization | True |
|  | Hidden units of forward and reverse model | 512 |
|  | Hidden units of encoder | $512 \times 256 \times 128$ |
| TSRL Architecture | Critic neural network layer width | 512 |
|  | Actor neural network layer width | 512 |
|  | State normalization | True |
|  | Actor learning rate | $3 \times 10^{-4}$ |
|  | Critic learning rate | $3 \times 10^{-4}$ |
|  | Policy noise | 0.2 |
|  | Policy noise clipping | 0.5 |
|  | Policy update frequency | 2 |
|  | Discount factor $\gamma$ | 0.99 |
|  | Number of iterations | $10^6$ |
|  | Target update rate | 0.005 |
|  | $\lambda_{L1}$ | 1e-5 |
| TSRL Hyperparameters | $\alpha$ | 2.5 |
|  | $\tau$ | 50% for Walker2d and Adroit tasks, 70% for HalfCheetah and Hopper2d |
|  | $\lambda_1$ | MuJoCo: 5 or 10 for full dataset, 100 or 200 for 10k dataset Adroit: 10,000 for both full and reduced datasets |
|  | $\lambda_2$ | 1 for MuJoCo full & Adroit datasets 100 for MuJoCo 10k dataset |

## B  Detailed Experiment Setups

**Reduced-size dataset generation.** To create reasonable reduced-size D4RL datasets for a fair comparison, we sub-sample the trajectories in the datasets rather than directly sampling the $(s, a, s', r)$ transitions. For example, there are 2M $(s, a, s', r)$ transitions in the "halfcheetah-medium-expert" dataset, we first split these records into 2,000 trajectories based on the done condition, then randomly draw 10 trajectories (10k transition points) to serve as the reduced-size datasets for model training.

**Experiment setups for representation learning evaluation.** To evaluate the representation quality and the impact of each design choice of TDM, we compare TDM representation with several baselines on the small dataset settings. We provide the detailed description of the representation learning baselines as follows:

- **"AE-rep" model:** We construct a vanilla auto-encoder without any further constraints during the learning process, which was trained by the reconstruction loss only. The network sizes of the encoder and decoders are the same as the ones used in TDM.

- **"AE-fwd-rep" model:** Similar to the "AE-rep" model but with a latent forward dynamics prediction model $f$, which is implemented as a 4-layer feed-forward neural network with ReLU activation, and optimized using Adam optimizer (same as TDM). The forward model was trained by minimizing the loss term $\|\dot{z}_s - f(\phi(s,a))\|_2^2$, where we directly regress $f(\phi(s,a))$ with the $\dot{z}_s$ derived from the latent states obtained from the encoder as $\dot{z}_s = z_{s'} - z_s$. Note that in this baseline, no ODE property nor T-symmetry regularization is included. Again we use the decoder to decode $\dot{z}_s \to \dot{s}$ as in TDM for the next state prediction.

- **"TDM-no-ODE" model:** Holds the same structure with TDM but trained with no ODE property. More specifically, similar with "AE-fwd-rep", the latent forward and reverse dynamics model was trained by $\|\dot{z}_s - f(\phi(s,a))\|_2^2$ and $\|(-\dot{z}_s) - g(\phi(s',a))\|_2^2$, where $-\dot{z}_s$ is directly calculated from the encoded latent states, i.e., $\dot{z}_s = z_{s'} - z_s$. Note that in this baseline, the T-symmetry is also implicitly captured, since both the latent forward and reverse dynamics models are regressing the same $\dot{z}_s$ and its opposite value.

- **"SimSiam" model:** For the self-supervised representation learning baseline, we implement an auto-encoder structure with the optimization objective proposed in the SimSiam paper [44]. For detailed model description and hyperparameters setting, please refer to Chen et al. [44].

**Experiment setups for evaluating generalization performance.** To evaluate TSRL's generalization capability beyond the offline datasets, we construct two low-speed datasets based on the original D4RL Walker2d medium and medium-expert datasets. In accordance with the Gym documentation, we selected the "x-coordinate velocity of the top" (8th dimension of the states) in the walker environment to perform data filtering. We remove all samples with the x-coordinate velocity of the top greater than $0.2\times$ max-x-velocity recorded in the data. This results in two smaller low-speed datasets (about 200k for the medium dataset and 250k for the medium-expert dataset). We train TDM and TSRL on these low-speed datasets and the results are reported in Figure 7 (main paper).

## C   Additional Results

**Complete results on D4RL Adroit tasks.** The complete results of TSRL in Adroit human and cloned tasks with different dataset scales are presented in Table 4. As shown in the results, TSRL achieves much better performance in the pen tasks, both the full datasets and the reduced-size datasets.

Table 4: Complete results on D4RL Adroit tasks

| Task | Ratio | Size | BC | TD3+BC | MOPO | CQL | IQL | DOGE | TSRL |
|---|---|---|---|---|---|---|---|---|---|
| Pen-human | 1 | 5k | 34.4 | 8.4 | 9.7 | 37.5 | 71.5 | $42.6 \pm 16.3$ | **80.1±18.1** |
| Hammer-human | 1 | 5k | 1.5 | 2.0 | 0.2 | **4.4** | 1.4 | $-1.2 \pm 0.2$ | $0.2\pm 0.3$ |
| door-human | 1 | 5k | 0.5 | 0.5 | -0.2 | **9.9** | 4.3 | $-1.1 \pm 0.2$ | $0.5\pm 0.3$ |
| Relocate-human | 1 | 5k | 0.0 | -0.3 | -0.2 | **0.2** | 0.1 | $-0.3 \pm 0.5$ | $0.1 \pm 0.1$ |
| Pen-cloned | 1 | 500k | 56.9 | 41.5 | -0.1 | 39.2 | 37.3 | $56.9 \pm 15.2$ | $\mathbf{64.9 \pm 20.1}$ |
| | 1/50 | 10k | $37.4 \pm 37.6$ | $-0.1 \pm 6.9$ | $-0.1 \pm 0.1$ | $1.5 \pm 4.8$ | $35.6 \pm 30.5$ | $30.1 \pm 19.7$ | $\mathbf{41.6 \pm 27.5}$ |
| Hammer-cloned | 1 | 500k | 0.8 | 0.8 | 0.2 | **2.1** | **2.1** | $0.2 \pm 0.3$ | $1.7 \pm 1.9$ |
| | 1/50 | 10k | $0.3 \pm 0.4$ | $0.2 \pm 0.1$ | $0.1 \pm 0.1$ | $0.2 \pm 0.1$ | $0.4 \pm 0.2$ | $0.3 \pm 0.1$ | $\mathbf{0.6 \pm 0.3}$ |
| Door-cloned | 1 | 500k | -0.1 | -0.4 | -0.1 | 0.4 | **1.6** | $-0.1 \pm 0.1$ | $-0.1 \pm 0.6$ |
| | 1/50 | 10k | $-0.1 \pm 0.1$ | $-0.3 \pm 0.1$ | $-0.2 \pm 0.1$ | $-0.3 \pm 0.1$ | $\mathbf{1.5 \pm 0.8}$ | $-0.5 \pm 0.5$ | $-0.1 \pm 0.3$ |
| Relocate-cloned | 1 | 500k | -0.1 | -0.3 | -0.3 | **0.1** | -0.2 | $-0.2 \pm 0.1$ | $-0.2 \pm 0.1$ |
| | 1/50 | 10k | $-0.2 \pm 0.1$ | $-0.3 \pm 0.1$ | $-0.3 \pm 0.1$ | $-0.3\pm 0.1$ | $\mathbf{-0.1 \pm 0.5}$ | $-0.2 \pm 0.1$ | $-0.2 \pm 0.1$ |

**Additional results on Antmaze-umaze tasks.** We also conduct experiments on the D4RL Antmaze-umaze tasks with full and reduced-size 10k datasets. The results are presented in Table 5. We use the

Table 5: Results on D4RL Antmaze-umaze tasks with full and reduced-size datasets

| Task | Ratio | Size | BC | TD3+BC | CQL | IQL | DOGE | TSRL(ours) |
|------|-------|------|----|--------|-----|-----|------|------------|
| Antmaze-u | 1 | 1M | 54.6 | 78.6 | 84.8 | 85.5 | **97.0 ± 1.8** | 81.4 ± 19.2 |
| | 1/100 | 10k | 44.7 ±42.1 | 0.7 ± 1.2 | 0.1 ± 0.0 | 65.1 ± 19.4 | 56.3 ± 24.4 | **76.1±15.6** |
| Antmaze-u-d | 1 | 1M | 45.6 | 71.4 | 43.4 | 66.7 | 63.5 ± 9.3 | **76.5 ± 29.7** |
| | 1/100 | 10k | 24.1±22.2 | 16.27 ± 16.4 | 0.5 ± 0.1 | 34.6 ± 18.5 | 41.7 ± 18.9 | **52.2 ±22.1** |

Table 6: Performance of data augmentation methods with 10k reduced-size D4RL datasets.

| Task | CABI | S4RL-N | S4RL-U | TSRL |
|------|------|--------|--------|------|
| Hopper-m | 48.3 ± 3.9 | 28.3 ± 6.2 | 23.6 ± 4.7 | **62.0 ± 3.7** |
| Hopper-mr | 19.8 ± 3.9 | 16.6 ± 12.9 | 12.5 ± 12.2 | **21.8 ± 8.2** |
| Hopper-me | 38.3 ± 5.3 | 12.5 ± 3.8 | 13.1 ± 4.7 | **50.9 ± 8.6** |
| Hopper-e | 34.6 ± 24.4 | 14.1 ± 12.9 | 12.2 ± 11.6 | **82.7 ± 21.9** |
| Halfcheetah-m | 34.8 ± 1.9 | 25.1 ± 6.8 | 23.2 ± 7.1 | **38.4 ± 3.1** |
| Halfcheetah-mr | 23.5 ± 3.4 | 15.1 ± 9.3 | 14.8 ± 9.5 | **28.1 ± 3.5** |
| Halfcheetah-me | 29.9 ± 1.7 | 27.1 ± 7.1 | 23.4 ± 8.2 | **39.9 ± 21.1** |
| Halfcheetah-e | 4.2 ± 4.1 | 2.4 ± 3.9 | 1.8 ± 3.1 | **40.6 ± 24.4** |
| Walker2d-m | 42.4 ± 23.3 | 24.5 ± 4.3 | 21.9 ± 4.8 | **49.7 ± 10.6** |
| Walker2d-mr | 11.7 ± 7.6 | 1.5 ± 2.1 | 1.4 ± 2.3 | **26.0 ± 11.3** |
| Walker2d-me | 17.4 ± 9.2 | 21.9 ± 16.4 | 16.0 ± 13.2 | **46.7 ± 17.4** |
| Walker2d-e | 20.2 ± 3.4 | 56.5 ± 26.7 | 51.1 ± 29.7 | **102.2 ± 11.3** |

same hyperparameters as in the D4RL MuJoCo tasks. Again, we find that TSRL achieves comparable performance as other baselines on the full datasets, but is substantially better under small datasets.

**Additional comparative results on data augmentation.** We conduct additional experiments on the reduced-size 10k MuJoCo datasets to compare TSRL and other offline RL methods using data augmentation. In particular, we compared with model-free method S4RL [42] with Gaussian (S4RL-N) and uniform (S4RL-U) noises, as well as a recent model-based data augmentation method CABI [43]. CABI employs a pair of predictive dynamics models to assess the reliability of the augmented data. The results are presented in Table 6.

The results clearly show that TSRL outperforms all offline RL baselines with data augmentation under small datasets. It is also observed that model-based methods TSRL and CABI generally perform better than model-free data augmentation method S4RL in this setting, due to access to additional dynamics information. Moreover, as CABI does not learn a strongly regularized dynamics model with T-symmetry consistency as in our proposed TDM, it still has a noticeable performance gap as compared to our method.

**Ablation on the level of ODE and T-symmetry regularization in TDM.** As discussed in the conclusion section of the main paper, TDM adds extra ODE dynamics and symmetry regularizations, which are beneficial to improve model generalization, but will lose some model expressiveness if the regularization is too strong. In this section, we conduct an ablation study on the impact of the regularization strength of the ODE property and the satisfaction with the T-symmetry. Specifically, we vary the loss weights of $\ell_{fwd}$, $\ell_{rvs}$ and $\ell_{T-sym}$ in the TDM learning objective (Eq. (8)), and train a loosely regularized and a strongly regularized TDM model on the 10k datasets (see Table 7). The loosely regularized model has the maximum reconstruction expressivity but may not produce a well-behaved representation due to weak regularization. Whereas the strongly regularized model sacrifices the expressivity for regularized behaviors. We further evaluate their performance with TSRL, with the results reported in Table 8. The experiment results demonstrated that an overly expressive model could not help the RL algorithm to derive a well-behaved policy with limited data

Table 7: TDM with different regularization strengths

| Different versions of TDM | $\ell_{rec}$ | $\ell_{ds}$ | $\ell_{fwd}$ | $\ell_{rvs}$ | $\ell_{T-sym}$ | $\lambda_{L1}$ |
|---|---|---|---|---|---|---|
| Loosely regularized | 1 | 1 | 0.01 | 0.01 | 0.01 | 1e-5 |
| Paper | 1 | 1 | 0.1 | 0.1 | 0.1 | 1e-5 |
| Strongly regularized | 1 | 1 | 1 | 1 | 1 | 1e-5 |

Table 8: Performance of TSRL with different TDM models on 10k datasets

| Task | TDM (loosely regularized) | TDM (paper) | TDM (strongly regularized) |
|---|---|---|---|
| Hopper-m | 50.7±13.6 | **62.0±3.7** | 43.6±14.3 |
| Hopper-m-r | 15.4±9.7 | **21.8±8.2** | 15.6±9.8 |
| Hopper-m-e | 49.7±17.1 | **50.9±8.6** | 30.9±20.5 |
| Halfcheetah-m | **39.1±3.6** | 38.4±3.1 | 36.6±30.0 |
| Halfcheetah-m-r | **28.3±6.9** | 28.1±3.5 | 22.9±8.4 |
| Halfcheetah-m-e | 36.2±5.4 | **39.9±21.1** | 31.0 ± 3.4 |
| Walker2d-m | 43.2±27.3 | **49.7±10.6** | 35.6±26.2 |
| Walker2d-m-r | 20.2±18.1 | **26.0±11.3** | 21.7±6.1 |
| Walker2d-m-e | 25.9±20.7 | **46.4±17.4** | 29.4±24.7 |

due to potential overfitting and inconsistency with the T-symmetry property. On the other hand, an overly regularized model may also hurt performance. This is consistent with our previous insight that a trade-off exists between model expressiveness and T-symmetry agreement. A proper balance between these two behaviors can be necessary for small-sample learning.

**Learning curves for TSRL on D4RL locomotion tasks.** The learning curves for reduced-size D4RL MuJoCo datasets with 10k samples are showed in Figure 8. For each evaluation step, the policies are evaluated with 5 episodes over 3 random seeds.

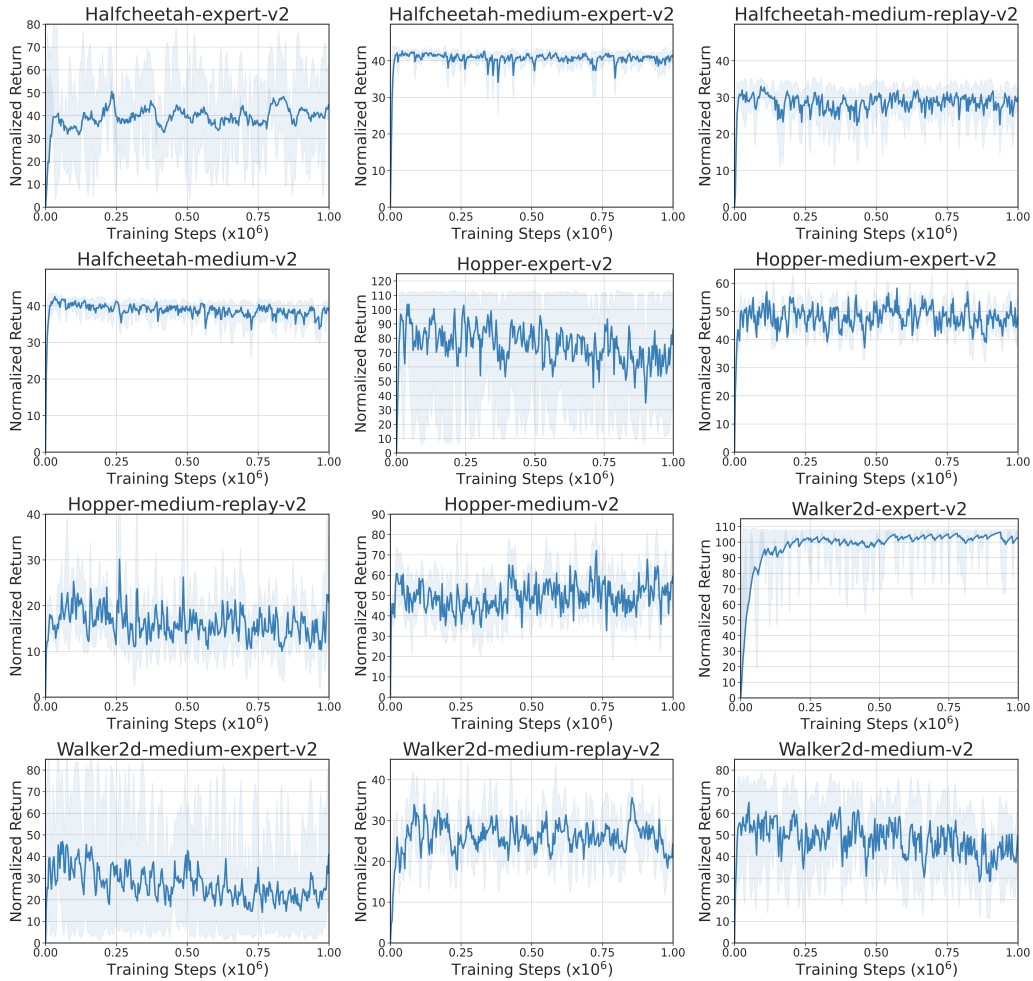

Figure 8: Learning curves for reduced-size D4RL MuJoCo datasets. Error bars indicate min and max values over 3 random seeds.

