# OpenReview forum: "Look Beneath the Surface: Exploiting Fundamental Symmetry for Sample-Efficient Offline RL"
_NeurIPS.cc/2023/Conference — NeurIPS 2023 poster_

### Official Review · Reviewer_xNUN · 2023-07-04

**Soundness:** 3 good
**Presentation:** 2 fair
**Contribution:** 2 fair
**Rating:** 5
**Confidence:** 3

**Summary:**

This paper proposes a physics-informed dynamics model TDM and a new offline RL algorithm TSRL, which exploit the fundamental symmetries in the system dynamics for sample-efficient offline policy learning, embedding and enforcing T-symmetry between a pair of latent forward, and reversing ODE dynamics to learn fundamental dynamics patterns in data. Empirical results on D4RL benchmark datasets validate the good generalization ability of TSRL.

**Strengths:**

- The idea is interesting and makes intuitive sense.
- The paper is overall well-written.
- Authors do comprehensive experiments on D4RL tasks to evaluate the generalization ability of the new method.

**Weaknesses:**

- The performance of TSRL is not comparable with other baselines in most Adroit human and cloned tasks.
- No comparison of other offline reinforcement learning methods using data augmentation.
- No experiments to validate TSRL alleviates the problem of over-conservatism.

**Questions:**

- Explain the reason that the performance of TSRL is not comparable with other baselines in most Adroit human and cloned tasks.
- Add comparisons of other offline reinforcement learning methods using data augmentation.
- Add related work of the data augmentation methods using offline RL in Sec. 6.

**Limitations:**

The limitations of the paper are properly addressed by the authors, but the societal impacts of the paper are not discussed.

---

> ### Author Rebuttal · Authors · 2023-08-09
>
> > **W1 & Q1. The performance of TSRL is not comparable with other baselines in most Adroit human and cloned tasks.**
>
> - The full results for Adroit tasks were listed in Appendix C Table 4 due to the space limit of the main article, please check our supplementary material for details. we observe that TSRL achieves much better performance as compared to the baseline algorithms in the pen tasks (both the full datasets and the reduced-size datasets), and comparable performance in other tasks. Note that Adroit tasks are substantially more challenging as compared to MuJoCo tasks due to their high-dimensionality and potentially non-Markovian (human dataset) property, most offline RL algorithms struggle in these tasks.
> - In Appendix C, we also provide the comparative performance of TSRL on Antmaze-umaze tasks with full and 10k reduced-size datasets. We find TSRL also achieves substantially better performance under small datasets of these tasks.
>
>
>
> > **W2 & Q2. No comparison of other offline reinforcement learning methods using data augmentation.**
>
> - In Fig. 5 and discussion in Section 5 L287-293, we have provided the comparison of TSRL with S4RL[1] (reference [42] in the paper) with Gaussian noises on two MuJoCo tasks.
> - To fully address the concern of the reviewer, we have conducted additional experiments on the reduced-size 10k MuJoCo datasets to compare TSRL and other offline RL methods using data augmentation. In particular, we compared with model-free method S4RL[1] with Gaussian (S4RL-N) and uniform (S4RL-U) noises, as well as a recent model-based data augmentation method CABI[2] (Reference [43] in the paper). CABI employs a pair of predictive dynamics models to assess the reliability of the augmented data. The KFC[3] (reference [32] in the paper) is also a model-based data augmentation method, which utilizes Koopman theory to model the dynamical system and augments data points in the latent linearizable space. However, KFC has no publicly available code, and re-implementing this algorithm is quite challenging, given the limited rebuttal period, we opt to evaluate CABI with our method. The detailed results are reported in the following table.
>
> **Table 1: Performance of data augmentation methods with 10k reduced-size D4RL datasets. Each result is generated by 3 random seeds.**
>
> | **D4RL tasks**| **CABI** |**S4RL-N**| **S4RL-U**| **TSRL**|
> | -------|----|----|----|----|
> |Hopper-m|48.3 $\pm$ 3.9| 28.3 $\pm$ 6.2| 23.6 $\pm$ 4.7| **62.0 $\pm$ 3.7**
> |Hopper-mr|19.8 $\pm$ 3.9|16.6 $\pm$ 12.9|12.5 $\pm$ 12.2|**21.8 $\pm$ 8.2**
> |Hopper-me|38.3 $\pm$ 5.3|12.5 $\pm$ 3.8| 13.1 $\pm$ 4.7|**50.9 $\pm$ 8.6**
> |Hopper-e|34.6 $\pm$ 24.4|14.1 $\pm$ 12.9|12.2 $\pm$ 11.6|**82.7 $\pm$ 21.9**
> |Halfcheetah-m| 34.8 $\pm$ 1.9|25.1 $\pm$ 6.8|23.2 $\pm$ 7.1|**38.4 $\pm$ 3.1**
> |Halfcheetah-mr| 23.5 $\pm$ 3.4|15.1 $\pm$ 9.3|14.8 $\pm$ 9.5|**28.1 $\pm$ 3.5**
> |Halfcheetah-me| 29.9 $\pm$ 1.7|27.1 $\pm$ 7.1|23.4 $\pm$ 8.2|**39.9 $\pm$ 21.1**
> |Halfcheetah-e| 4.2 $\pm$ 4.1|2.4 $\pm$ 3.9|1.8 $\pm$ 3.1|**40.6 $\pm$ 24.4**
> |Walker2d-m| 42.4 $\pm$ 23.3|24.5 $\pm$ 4.3|21.9 $\pm$ 4.8|**49.7 $\pm$ 10.6**
> |Walker2d-mr|11.7 $\pm$ 7.6|1.5 $\pm$ 2.1|1.4 $\pm$ 2.3|**26.0 $\pm$ 11.3**
> |Walker2d-me|17.4 $\pm$ 9.2|21.9 $\pm$ 16.4|16.0 $\pm$ 13.2|**46.7 $\pm$ 17.4**
> |Walker2d-e|20.2 $\pm$ 3.4|56.5 $\pm$ 26.7| 51.1 $\pm$ 29.7|**102.2 $\pm$ 11.3**
>
>
> - The results clearly show that TSRL outperforms all offline RL baselines with data augmentation under small datasets. It is also observed that model-based methods TSRL and CABI generally perform better than model-free data augmentation method S4RL in this setting, due to access to additional dynamics information. Moreover, as CABI does not learn a strongly regularized dynamics model with T-symmetry consistency as in our proposed TDM, it still has a noticeable performance gap as compared to our method.
>
>
> > **W3. No experiments to validate TSRL alleviates the problem of over-conservatism.**
>
> - We have conducted experiments to evaluate the OOD generalization performance of TSRL, which is direct evidence to show TSRL can alleviate the problem of over-conservatism. If the reviewer check Fig. 7, L307-319 in Section 5 as well as Fig. 8, L592-607 in Appendix C, we constructed two low-speed datasets from D4RL-Walker2d datasets by removing all high x-velocity samples. This creates smaller datasets with a large proportion of state-action space and transition dynamics unobserved. As the rewards in these tasks encourage high-speed behavior, we want to test if the agent can still generalize and learn well given only these low-speed data.
> - As shown in Fig.7 and 8, the existing offline RL algorithms perform poorly when trained with only low-speed data, which is primarily due to the adoption of over-conservative data-related regularizations, making these algorithms not able to deviate largely from the data distribution. By contrast, TSRL is still able to achieve good performance, due to the access to more fundamental dynamics information that remains invariant in both low- and high-speed data. This is evident if the reviewer inspects the policy rollout distributions (right figures of Fig. 7 and 8) that the policies learned by TSRL indeed generalize to novel high-speed behaviors that are not present in the training data.
>
>
> > **Q3. Add related work of the data augmentation methods using offline RL in Sec. 6.**
>
> - We thank the reviewer's suggestion. We will add more detailed disussion on related work and include the additional experiment results in our final paper.
>
>
>
> ## References
> [1] Sinha, S., Mandlekar, A., & Garg, A. S4rl: Surprisingly simple self-supervision for offline reinforcement learning in robotics. CoRL 2022.
>
> [2] Lyu, J., Li, X., & Lu, Z. Double Check Your State Before Trusting It: Confidence-Aware Bidirectional Offline Model-Based Imagination. NeurIPS 2022.
>
> [3] Weissenbacher, M., et al., Koopman q learning: Offline reinforcement learning via symmetries of dynamics. ICML 2022.

---

> > ### Comment · Area_Chair_LfMZ · 2023-08-21
> > **Reviewer, please respond to the rebuttal**
> >
> > Reviewer, please respond to the rebuttal.

---

### Official Review · Reviewer_dZPC · 2023-07-05

**Soundness:** 2 fair
**Presentation:** 3 good
**Contribution:** 2 fair
**Rating:** 5
**Confidence:** 4

**Summary:**

The current offline RL algorithm requires a large amount of offline dataset training and has poor performance on small datasets. This article proposes a framework to address this issue. By learning a T-symmetry enhanced dynamic model, capture more fundamental dynamic relationships. Afterward, the article applies T-symmetry to offline RL, uses a model to regularize constraints in the latent action space, and uses a model to filter data for data augmentation. The experimental results showed improvement in small datasets.

**Strengths:**

1. The key idea is novelty. This paper explores symmetries to enhance the performance of offline RL with small datasets. The proposed method has a new technical insight.

2. The article is clearly structured.

3. The proposed method improves performance on offline RL with small datasets.


**Weaknesses:**

1. This article chooses the backward model because of irreversible action, but this does not quite meet the definition of time-reversal symmetry. Other articles (https://arxiv.org/abs/2111.12600) respectively choose the inverse model to get the reversible action, and then process the irreversible action, rather than simply giving up.

2. The proposed method has a great improvement on small datasets, but there is still a lot of gap from the performance of complete data.


**Questions:**

1. Compared with other methods, since t-symmetry can learn better representation and dynamics, why is there a great improvement in small datasets, but not in full datasets?

2. Why choose to perturb the latent space and then filter the data, rather than perturbing directly on the state space, and then filtering? Is this part of the performance improvement caused by the latent space, the filtered data, or both?


**Limitations:**

Yes

---

> ### Author Rebuttal · Authors · 2023-08-08
>
> > **W1. Difference with the origianl definition of time-reversal symmetry. Comparison with the treatment on irreversible actions in https://arxiv.org/abs/2111.12600.**
>
> We thank the reviewer for providing this reference and will add it to our final paper. Regarding the differences:
>
> - As we have discussed in the last subsection in Section 2 (L113-129), the original T-symmetry (i.e., $F(s,a)=\dot{s}=-\tilde{G}(s',a')$) can sometimes be broken by irreversible action $a'$. Hence we consider an extended form, that is, enforcing $F(s,a)=\dot{s}=-G(s',a)$ while preserving the first-order ODE requirement in T-symmetry. Note that this condition is almost universally held in typical MDP, as it essentially suggests the distribution $P(s,a,s')$ and $P(s',a,s)$ should be equal. It also has the advantage of being irrelevant to the impact of irreversible actions $a'$. Furthermore, using the reverse dynamics model (e.g., function mapping with $(s',a)\rightarrow s$) has already been adopted in many RL studies [1,2,3], our method differs from these works in that we use it to construct T-symmetry consistency and models it as an ODE system.
> - The CCWM (Cycle-Consistency World Model) mentioned by the reviewer actually adopts a more engineering-oriented approach to address the issue of irreversible actions. It removes "irreversible" transitions that produce sudden changes in Q-values over a trajectory in the latent space from the modeling process. This actually ignored some dynamics properties in an MDP associated with such irreversible actions. While in our work, such "irreversible" transitions will not cause a problem and can be implicitly captured in our modeling process.
>
>
> > **W2 & Q1. The proposed method has a great improvement on small datasets, but there is still a lot of gap from the performance of complete data.**
>
> - As we have discussed in the conclusion, there is a trade-off between model generalizability and expressiveness. In TDM, we introduce multiple regularizers as well as the additional T-symmetry regularization to obtain a well-behaved dynamics model for the small-dataset setting. But this could hurt model expressiveness, as the model tries to extract the fundamental representations/patterns within the dataset, rather than fits each individual sample. This is beneficial for improving robustness and generalization in the low-data regime but can be overly restrictive for large datasets.
> - In offline RL, learning from very small datasets and large datasets faces different challenges and potentially requires different algorithm design logics. For large datasets with high state-action space coverage, maximally exploiting the offline dataset and ensuring policy learning within data distribution to avoid distributional shift is already sufficient, which is exactly how most existing offline RL algorithms are designed. While under small datasets, strictly regularizing policy within data distribution will hurt performance, encouraging OOD generalization is the key to achieving higher performance. The data scarcity in the small-sample setting makes it more challenging to learn a reliable policy, thus requiring robust regularization techniques to prevent overfitting and promote generalization performance.
> - Finally, it should be noted that although our proposed TSRL is designed for the small-dataset setting, as shown in Table 1 and Fig. 3, it can still achieve comparable (sometimes even better) performance with existing advanced offline RL methods in the complete datasets, despite the fact that D4RL provides overly large amounts of data for simple locomotion tasks.
>
>
> > **Q2. Why choose to perturb the latent space and then filter the data, rather than perturbing directly on the state space, and then filtering?**
>
> - First, in our TDM, the T-symmetry consistency is enforced on the latent forward and reverse dynamics rather than on the original state-action space. Thus it has to evaluate T-symmetry violations in the latent space.
> - The forward and reverse dynamics are learned in latent space because they need to be first-order ODE systems to establish the T-symmetry relationship. For a general nonlinear dynamical system, directly fitting a first-order ODE system can suffer from relatively large errors. Hence the current best practices in the control community (such as Koopman theory [4,5] and Sindy [6,7]) first map the original state-action space into a well-behaved latent space and then construct latent first-order ODE dynamics by fitting the data.
> - Lastly, perturbing in the latent space for data augmentation is much more convenient for implementation. As in our TSRL, the inputs of the Q-function are the latent states and actions ($(z_s, z_a)=\phi(s,a)$) from the TDM encoder $\phi$. And the augmented data is executed in the latent space while training the Q function. Perturbing directly on the original state space will incur another origin-to-latent conversion step and cause extra computation.
>
>
> ## References
> [1] Lai, H. et al., Bidirectional model-based policy optimization. ICML 2020.
>
> [2] Wang, J. et al., Offline reinforcement learning with reverse model-based imagination. NeurIPS 2021.
>
> [3] Lyu, J., Li, X., & Lu, Z. Double Check Your State Before Trusting It: Confidence-Aware Bidirectional Offline Model-Based Imagination. NeurIPS 2022.
>
> [4] Weissenbacher, M., et al., Koopman qlearning: Offline reinforcement learning via symmetries of dynamics. ICML 2022.
>
> [5] Mezic, I. Spectral properties of dynamical systems, model reduction and decompositions. Nonlinear Dynamics, 2005.
>
> [6] Brunton, S., et al., Discovering governing equations from data by sparse identification of nonlinear dynamical systems. PNAS, 2016.
>
> [7] Champion, K., et al., Data-driven discovery of coordinates and governing equations. PNAS, 2019.

---

> > ### Comment · Reviewer_dZPC · 2023-08-18
> >
> > Thank you very much for the responses which address and answer my questions. After reading the rebuttal and reviews of other reviewers, I decide to keep my score.

---

### Official Review · Reviewer_5ovg · 2023-07-06

**Soundness:** 3 good
**Presentation:** 3 good
**Contribution:** 3 good
**Rating:** 5
**Confidence:** 4

**Summary:**

This work introduced a Time-reversal symmetry enforced dynamics model, which leverages the consistency between a pair of forward and reverse latent dynamics for improving the sample efficiency of offline RL algorithms. Conducted experiments demonstrate the effectiveness of the proposed method.

**Strengths:**

- The proposed method makes sense in improving the sample efficiency of offline RL algorithms.
- This work is clearly presented so that it is easy for readers to catch up with the main ideas.

**Weaknesses:**

- I suggest authors to strengthen the analysis on the rationale, i.e., why can the proposed method improve the sample efficiency of off-line RL algorithms.  This will make the paper more insightful.
- This work lacks sufficient discussion on the relation/comparison with related works, especially for those also exploiting the consistency, e.g.,    PlayVirtual [1] and some others cited by [1].
[1] Yu, Tao, et al. "Playvirtual: Augmenting cycle-consistent virtual trajectories for reinforcement learning." Advances in Neural Information Processing Systems 34 (2021): 5276-5289.


**Questions:**

1. Why can the proposed method improve the sample efficiency of off-line RL algorithms? Pls give in-depth and convincing analysis.
2. What are the relations between this work and those metioned in the weakness part?
3.An open question: Are there any ideas to extend the core idea of this work to Online RL algorithms?

**Limitations:**

Pls see the weakness and questions parts.

---

> ### Author Rebuttal · Authors · 2023-08-08
>
> > **W1 & Q1. Rational of the sample efficiency improvement of TSRL**
>
> The sample efficiency of TSRL is a joint result of a series of elegant and closely related design choices:
>
> - Firstly, learning fundamental/parsimonious dynamics is essential to improve model performance under small datasets. Learning fundamental properties within data, in principle, does not require large or high-coverage datasets. Moreover, a fundamental model can also help remove spurious correlations and maximally promote model stability and generalization under limited data (see the discussion in L54-65 in our paper).
> - T-symmetry happens to be one of the simplest and most fundamental properties that we can leverage to enforce such a fundamental property (see the discussion in L54-65, L104-129). We further extend T-symmetry to make it broadly applicable to generic MDP settings (see the discussion in L113-129).
> - By modeling the latent dynamics to be first-order ODE systems (forcing the latent dynamics to be mathematically simple) and enforcing extended T-symmetry, we can obtain a well-regularized dynamics model (TDM). This model can provide a more effective state-action representation for offline RL (see the discussion in L184-199 and our ablation on representation learning in Fig. 4).
> - The deviation in latent actions and the consistency with T-symmetry specified in TDM actually provide another perspective to detect unreliable or non-generalizable samples. These deviations can serve as a new set of policy constraints to replace the highly restrictive OOD regularizations in existing offline RL algorithms (see the discussion in L200-207, 216-222). Moreover, as we have shown in Fig. 7, this new type of policy constraint allows the policy to reliably generalize to OOD regions without being constrained to the training data distribution, leading to better small-sample performance (see L307-319 for detailed discussion).
> - Lastly, compliance with T-symmetry also enables a reliable latent data augmentation scheme that further addresses the limited size of training data (see empirical analysis in L287-293).
>
>
> > **W2 & Q2. Insufficient discussion on the related works, especially PlayVirtual [1].**
>
> We thank the reviewer for providing this reference, and we will add it to our final paper. Regarding the relationship and difference between TSRL and PlayVirtual:
> - **Problem setting and model construction:** PlayVirtual is designed for online RL settings where the dynamic models can be continuously improved with fresh interaction data to mitigate multi-step rollout compounding errors. While in offline small dataset settings, multi-step rollouts can lead to considerable compounding errors, even adding cycle-level forward & backward consistency may not be sufficient to properly regulate the model.
> On the other hand, our proposed TDM **does not perform any rollout generation**. It directly regulates the T-symmetry consistency between ODE latent forward and reverse dynamics at each step ($F(s,a) = -G(s',a)$), which applies a much stronger regulation to improve the model's small-sample performance.
> - **Model learning:** PlayVirtual learns the forward and backward dynamics models with the supervision of the future/previous state representations. While in our TDM, the latent forward and reverse dynamic model are learned as first-order ODEs ($F(s,a)={dz_s \over dt}$ ; $G(s',a)={-dz_s \over dt}$ ) to extract the essential patterns within the dynamical system.
> - **Integration with RL algorithm:** PlayVirtual directly incorporate the learned dynamics with standard online RL algorithm (e.g., SAC and Rainbow). While in our paper, we propose a new offline RL algorithm TSRL that closely integrates the learned TDM, allowing the full use of the dynamics-enhanced information from TDM to improve offline RL performance under limited data.
>
>
> > **Q3. Are there any ideas to extend the core idea of this work to online RL algorithms?**
>
> We thank the reviewer for this thoughtful comment. In principle, our method is also applicable to the online setting, but there are several modifications that could be introduced to our method to enable the best performance:
> - A small set of initial samples need to be collected in the environment using an initial or random policy to warm start the learning of TDM. This will make the learned representation evolve smoothly during the early RL training stage.
> - The strong regularizations in TDM could be properly relaxed, as online samples can be used to continuously improve model learning. We can trade off some regularization to promote model expressiveness.
> - The offline backbone RL algorithm in TSRL needs to be changed to an online RL algorithm with some exploration schemes to remove the pessimism. The T-symmetry regularized policy constraints are probably not necessary for the online setting and can be removed.

---

> > ### Comment · Reviewer_5ovg · 2023-08-21
> > **Thanks for your responses.**
> >
> > I confirm that I read your responses, and suggest you to add the contents in your rebuttal to the revised paper.

---

> > ### Comment · Area_Chair_LfMZ · 2023-08-21
> > **Reviewer, please submit your response to author's rebuttal**
> >
> > Please read the rebuttal of the authors and respond.

---

### Official Review · Reviewer_P7X7 · 2023-07-31

**Soundness:** 3 good
**Presentation:** 3 good
**Contribution:** 3 good
**Rating:** 7
**Confidence:** 4

**Summary:**

The paper investigates the time-reversal symmetry of forward and reverse dynamics in reinforcement learning (RL). The authors propose a Time-reversal symmetry enforced Dynamics Model (TDM) that models the consistency between forward and reverse dynamics. Using the TDM, they further propose an offline RL algorithm that leverages the learned TDM in three ways: 1. Using the representation from TDM for value function learning; 2. Using TDM to penalize OOD samples; 3. Using TDM to moderate useful data augmentation. Extensive experiments demonstrate that the proposed method outperforms a number of baselines, especially, the proposed method can learn a better policy with significantly fewer samples.

**Strengths:**

1. The proposed method is novel and backed by convincing experimental results.
2. The concept of time-reversal symmetry could potentially be widely applied in RL, as it represents a fundamental structure in many RL problems.


**Weaknesses:**

1. The proposed method leverages a learned dynamic model for offline RL, whereas most of the baselines are model-free (except for MOPO), which seems somewhat unfair. In particular, the proposed method appears somewhat related to the Dreamer [A, B] method, which also leverages a dynamic model for data augmentation. I am curious about the authors' thoughts on comparing with Dreamer.
2. The claim of leveraging time-reversal symmetry is slightly concerning as the method essentially learns the forward and reverse dynamic models instead of leveraging the time-reversal symmetric property of the MDP (e.g., P(s, a, s')=P(s', a, s) in certain scenarios).

[A] Hafner, Danijar, et al. "Dream to Control: Learning Behaviors by Latent Imagination." International Conference on Learning Representations. 2019.
[B] Hafner, Danijar, et al. "Mastering Atari with Discrete World Models." International Conference on Learning Representations. 2020.

**Questions:**

1. I wonder how the authors ensure the fairness of their experiments, given that the proposed method appears more complex than the baselines. Do all the baselines have a similar amount of trainable parameters as the proposed method?
2. Instead of ensuring the T-symmetry using an extra loss term, is it possible to constrain the neural network architecture so that the model respects the T-symmetry by definition (like an equivariant neural network architecture)?
3. The proposed method doesn't seem to be specific to offline RL. Have the authors tried to use it in online RL?

**Limitations:**

The authors do address the limitations of their work. However, the discussion could be more comprehensive. For example, under what circumstances would the T-symmetry break? Are there scenarios where the proposed method might fail?

---

> ### Author Rebuttal · Authors · 2023-08-08
>
> We really appreciate the reviewer for the positive feedback and valuable comments.
>
> > **W1. Comparison mostly to model-free methods rather than model-based methods. In particular, the comparison with Dreamer.**
>
> - First, we'd like to highlight that our approach is very different from the typical model-based methods. Most existing model-based methods use the learned dynamics model to generate imaginary rollouts to facilitate policy learning. Whereas in our approach, we **do not** use the model for rollouts generation, but rather only used it to learn well-behaved representations as well as consistency metrics for policy constraints and data augmentation. If the reviewer inspects our value and policy learning procedure in Section 4. it actually shares more similarities with many model-free policy constraint offline RL methods.
> - We adopt such a design as under offline small dataset settings, it is generally not possible to learn an accurate dynamics model for reliable rollout generation. A poor dynamics model can negatively impact policy learning. However, learning reasonable representations and consistency check metrics still remains possible if we learn a more fundamental dynamics model with strong physics-informed regularizations. This is evident as shown in Fig. 1, 3, and Table 1 in our paper, that model-based offline RL method like MOPO suffers from severe performance degradation with reduced data size as compared to model-free methods and our proposed TSRL.
> - Lastly, Dreamer v1&v2 are online RL methods, which allow continually acquiring fresh environment samples to improve the accuracy of the model. While in our setting, only a very small offline dataset is given. This significantly exacerbates the difficulty of model and policy learning. Note that even being sample-efficient, in the Dreamer paper, it still needs $5\times 10^6$ online environment samples to learn good policies for many tasks, while in our setting, we only provide each algorithm 10k offline samples.
>
>
> > **W2. Not leveraging the time-reversal symmetric property of the MDP (e.g., P(s, a, s')=P(s', a, s) in certain scenarios)**
>
> - If the reviewer closely inspects our proposed extended T-symmetry, we introduce two ODE systems $F(s,a) = \dot{s}$ and $G(s',a)=-\dot{s}$, where $\dot{s}=s'-s$. This is equivalent to constructing two consistent systems of $(s,a)\rightarrow s'$ and $(s',a)\rightarrow s$, essentially similar to the $P(s, a, s')=P(s', a, s)$ relationship in MDP mentioned by the reviewer, with the difference that we further require both dynamics are ODE systems.
> - Due to the above construction, our proposed extended T-symmetry is more broadly applicable as compared to the original T-symmetry (i.e., $F(s,a) = \dot{s}=-G(s',a')$) in MDP settings, as the latter can sometimes be broken by irreversible actions. Please refer to L113-129 in our paper for a detailed discussion.
>
> > **Q1. Fairness regarding using more complex model. Do all the baselines have a similar amount of trainable parameters as the proposed method?**
>
> - In this paper, we propose an RL algorithm rather than a supervised learning model, the setting is different and the number of parameters does not necessarily determine the performance. Note in the Dreamer v1&v2 papers, these more complex methods are also compared with lightweight methods like DQN and A3C.
> - To ensure comparability, we use the same architectures for Q-networks and policy network as other baseline algorithms in our experiments. The only addition is the incorporation of TDM for representation learning and consistency metric computation. Note that simply adding extra model/parameters in representation learning may not yield comparable performance improvements as in TSRL. As reported in Fig. 4 of our paper, baselines using SimSiam and AE-fwd-rep representations have a similar amount of parameters as in TSRL, but their performance improvements are less significant.
>
> > **Q2. Possibility of constraining the neural network architecture to enforce T-symmetry by definition (e.g. equivariant NN)?**
>
> - We appreciate the reviewer for the insightful comment. We actually explored the possibility of employing equivariant NN architectures to enforce T-symmetry, but we found some applicability issues:
>     - Most existing equivariant NN techniques leverage relatively simple and explicitly known equivariant mapping/transformation of the system. However, in our problem, the forward and reverse ODE dynamics are unknown and need to be learned. Enforcing equivariant relationships between two unknown systems is less straightforward to be implemented in a NN architecture.
> - In contrast, our method embraces simplicity and generality, without relying on excessive data or task-specific knowledge. We enforce T-symmetry by simply incorporating a few supervised loss terms. Nevertheless, constraining the NN architecture is a valuable idea, we will continue to explore this direction in our future works.
>
>
> > **Q3. Applicability to online RL?**
> - In principle, our method can also be used in online RL. However, as discussed in W1, online RL allows continuous collection of fresh samples through online interaction, hence the accuracy of the model can be improved during training. This makes heavy regularization used in TDM less necessary for online settings, especially in the later part of the training.
> - We focus on the small-sample offline RL setting, as it poses some more demanding challenges, and has many real-world deployment scenarios. Only a very small number of samples are given, one has to incorporate strong regularization to ensure reasonable generalization as well as alleviate distributional shift in offline policy optimization.
>
>
> > **Limitations: more discussion on the cases when T-symmetry breaks**
>
> - Please refer to our response to W2. Our proposed extended T-symmetry improves over the original T-symmetry definition for generic MDP settings and is not susceptible to the negative impact of irreversible actions.

---

> > ### Comment · Reviewer_P7X7 · 2023-08-11
> >
> > The reviewer thanks the authors for their thoughtful rebuttal. Most of my concerns are addressed, but I would like to clarify on W2.
> >
> > I apologize for mistyping the equation in my review, what I meant was $P(s, a, s')=P(s', \mathbf{-a}, s)$, which is similar to the original T-symmetry $F(s, a)=\dot{s}=-G(s', a')$. I understand that the proposed approach is not limited to irreversible actions, but the original T-symmetry is. However, would the original T-symmetry be more efficient when the assumption of action being reversible is satisfied? Essentially, the original T-symmetry has the potential to automatically generalize to reversed transitions, but the proposed method cannot.
> >
> > This is by no means criticizing the proposed approach, instead, I acknowledge that the proposed method resolves the reversible assumption. My point is that, first, as mentioned above, though the original T-symmetry is more constrained, does it have advantages when the assumption is satisfied? Second, IMO the word `T-symmetry` implies the original T-symmetry where the transition is symmetric when the time is reversed. I am slightly concerned about the wrong implications it might have for the readers.

---

> > > ### Author Response · Authors · 2023-08-13
> > > **Response to the Reviewer P7X7 Comments (1/2)**
> > >
> > > We really appreciate the reviewer for the thoughtful comment. Regarding the new comments on T-symmetry:
> > > - The original T-symmetry for physical systems is actually defined on state measures $\mathbf{x}\in\Omega$ of dynamical systems (i.e., $d \Gamma(x)/dt=-F(\Gamma(x))$, $\Gamma$ is the time invertible transformation), rather than state-action pairs $(s,a)$ for typical control problems. In these systems, the evolution on $\mathbf{x}$ is determined by some underlying physical laws. While for control problem, we have an external control policy to influence state evolution. Hence there has to make some adaptation to the original T-symmetry in order to make it usable in control problems, especially for the MDP setting.
> > > - In our adapted *extended T-symmetry* for the MDP setting, we preserved most of the characteristics of the original T-symmetry definition, such as using ODE forward and reverse dynamics, and time-reversal on states (i.e., $F(s,a)=\dot{s}=-G(s',a)$). We model the reverse dynamics as $G(s',a)$ rather than $G(s',a')$ as it can overcome the irrevsible action issue while still roughly following the core idea of T-symmetry.
> > > - Moreover, in our abstract, introduction, preliminaries as well as method sections in our paper, we only refer to our treatment as the "extended T-symmetry" rather than "T-symmetry" to demonstrate that it is an adaptation on the original T-symmetry.
> > > - Finally, in the early development stage of our work, we actually tested using $G(s',a')$ as the latent reverse dynamics, but it leads to inferior performance as compared to our final form $G(s',a)$, probably due to the inability to capture the irreversible actions. We will reproduce these results and reply in a follow-up post in the next 2 days. We hope this can address your remaining concerns about our method.

---

> > > > ### Author Response · Authors · 2023-08-13
> > > > **Response to the Reviewer P7X7 Comments (2/2)**
> > > >
> > > > Dear reviewer, we have finished the experiments by only changing the latent ODE reverse dynamics $g(z_{s'},z_a)$ to $g(z_{s'},z_{a'})$ in TDM (i.e., using $G(s',a')$ rather than $G(s',a)$), all other configurations in the TDM model are kept the same. The final performance of TSRL on the 10k D4RL reduced datasets are provided in the following table:
> > > >
> > > > **Table 1: Performance of TSRL with different reverse dynamic models with 10k data, each result is generated by 3 random seeds**
> > > >
> > > > | **D4RL tasks**| **$g(z_{s'},z_{a'})$**| **$g(z_{s'},z_a)$**|
> > > > | -------|:--------:|:--------:|
> > > > |Hopper-m| 50.5 $\pm$ 11.6| **62.0 $\pm$ 3.7**
> > > > |Hopper-mr|11.4 $\pm$ 7.3|**21.8 $\pm$ 8.2**
> > > > |Hopper-me|45.8 $\pm$ 10.7|**50.9 $\pm$ 8.6**
> > > > |Halfcheetah-m|36.3 $\pm$ 5.8|**38.4 $\pm$ 3.1**
> > > > |Halfcheetah-mr| 26.2 $\pm$ 7.7|**28.1 $\pm$ 3.5**
> > > > |Halfcheetah-me| 37.6 $\pm$ 4.8|**39.9 $\pm$ 21.1**
> > > > |Walker2d-m|38.1 $\pm$ 7.5|**49.7 $\pm$ 10.6**
> > > > |Walker2d-mr|20.8 $\pm$ 16.2|**26.0 $\pm$ 11.3**
> > > > |Walker2d-me|25.2 $\pm$ 20.9|**46.7 $\pm$ 17.4**
> > > >
> > > > - As we can observe in the above results, using $G(s',a')$ will lead to inferior performances in all tasks. We think the performance degradation may be due to the following reasons:
> > > >     - As we have described in our previous discussion, using $G(s',a')$ cannot capture the irreversible actions. Our design of using $G(s',a)$ on the other hand provides a much more generic form for the MDP settings.
> > > >     - Second, if we use both $a$ and $a'$ to train a dynamics model with the forward and reverse dynamics $F(s,a)$ and $G(s',a')$, as $a'$ is from the behavior policy $\pi_{\beta}(s')$ in the dataset, then the learned dynamics model will implicitly capture the characteristics of the behavior policy. This is not desirable, as we want the dynamics model to only capture fundamental system transition dynamics $P(s'|s,a)$. Including information related to the behavior policy will introduce undesirable spurious correlations between $s'$ and $a'$. By contrast, if we model TDM only using $F(s,a)$ and $G(s',a)$, then it basically uses the same information as in the transition dynamics $P(s'|s,a)$, which can help learn a more fundamental dynamics model.

---

> > > > > ### Comment · Reviewer_P7X7 · 2023-08-15
> > > > >
> > > > > Thank you for the discussion and the new experiment. My concerns are resolved and I would like to increase my evaluation to 7.
> > > > >
> > > > > The new experiments clearly shows that the extended T-symmetry outperforms the original T-symmetry, but I am curious that if the environment's action is fully reversible, would the extended T-symmetry still outperform? This is probably out of the scope of the paper, and I am just mentioning this for the purpose of discussion rather than proposing new experiments.
> > > > >
> > > > > Moreover, it just came to me that there is another paper that used the learned dynamic model to improve RL [A]. This is probably not super related because they were doing transfer learning, but I just feel that both works might share a high-level idea of using dynamic model to regularize RL.
> > > > >
> > > > > [A] Sun, Yanchao, et al. "Transfer RL across Observation Feature Spaces via Model-Based Regularization." International Conference on Learning Representations. 2021.

---

> > > > > > ### Author Response · Authors · 2023-08-16
> > > > > > **Thanks for raising the score**
> > > > > >
> > > > > > We want to express our sincere gratitude to the reviewer for increasing the score.
> > > > > >
> > > > > > In response to the comment, we will explore more task environments with different dynamical properties in the future to further examine the performance of our method. We appreciate the reference provided by the reviewer and will add it to our final paper.

---

### Decision · Program_Chairs · 2023-09-21

**Decision:**

Accept (poster)

**Comment:**

This paper explores the enhancement of dynamic model learning in offline RL by leveraging the symmetry of forward and reverse dynamics. The submission garnered predominantly favorable feedback. All reviewers concurred on the merit and novelty of the concept. While there were initial reservations regarding the robustness of the experiments, these concerns were substantially mitigated following the rebuttal and subsequent discussions. Consequently, the AC has decided to accept the submission.